# SimKO: Simple Pass@K Policy Optimization

## Abstract

Reinforcement learning with verifiable rewards (RLVR) has advanced the reasoning capabilities of large language models (LLMs). However, prevailing RLVR methods exhibit a systematic bias toward exploitation over exploration, as evidenced by improved pass@1 but reduced pass@K (K>1) performance. Such bias limits the advancement of LLMs' reasoning frontier. To understand this issue, we analyze training dynamics of RLVR methods by tracking the token-level probability distributions over vocabulary candidates. Our analysis reveals a consistent probability concentration effect where the top-1 candidate increasingly accumulates probability mass and suppresses that of other candidates. More importantly, stronger over-concentration correlates with worse pass@K performance. Inspired by this finding, we propose **Simple Pass@K Optimization (SimKO)**, a method designed to mitigate the over-concentration issue, thereby encouraging exploration. SimKO operates in an asymmetrical manner. For verified-correct responses, it boosts the probabilities of the top-K candidates. For verified-incorrect responses, it applies stronger penalties to the top-1 candidate. We observe that this asymmetric design is particularly effective at mitigating over-concentration when applied at tokens with high entropy. Across various math and logical-reasoning benchmarks, SimKO consistently yields higher pass@K for a wide range of K, providing a simple way to improve RLVR's exploration.

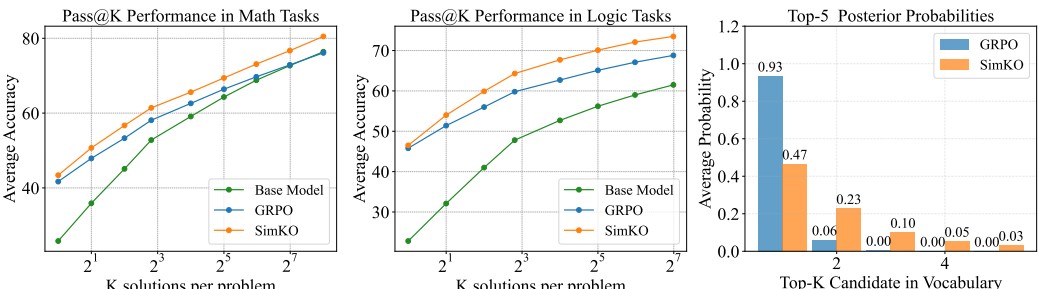

Figure 1: SimKO improves pass@K performance on math tasks (AIME24/25, AMC, MATH500, Minerva, Olympiadbench) and logic tasks (Synlogic, BBH) compared to GRPO, as shown in the plots (left and middle). The figure on the right shows the $k$-th highest candidate probabilities averaged over the dataset. The SimKO-trained model exhibits a less concentrated probability distribution compared to GRPO.

## 1 Introduction

Reinforcement Learning with Verifiable Rewards (RLVR) offers a simple recipe for improving LLM reasoning: the model generates responses, and updates itself by increasing the probability of the correct ones while decreasing that of the incorrect ones. This coupled process induces a systematic bias, whereby the model progressively collapses to a narrow set of safe responses, prioritizing exploitation over exploration. The effect is evident in improved pass@1, which measures the expected quality of a single reasoning path, but degraded pass@K, which measures the coverage of multiple reasoning paths (Yue et al., 2025; Wu et al., 2025). The reduced exploration ability limits the model's reasoning potential and deteriorates its ability to generalize to novel or more challenging scenarios.

Several approaches have been proposed to mitigate this exploration deficit. *Data-centric* methods focus on data augmentation, exposing the model to broader reasoning environments to promote

response diversity. Techniques include using off-policy data (Dong et al., 2025; Li et al., 2025), generating more diverse responses (Yang et al., 2025), and creating new task variations from the model itself (Liang et al., 2025). As a complementary line of work, *reward-centric* methods derive group-wise rewards to evaluate the collective quality of multiple responses. These approaches provide an unbiased estimate of pass@K (Walder & Karkhanis, 2025; Chen et al., 2025b), which the model can then directly optimize. Despite their promise, both families of methods operate only at the level of inputs or outputs, leaving the internal learning dynamics underexplored. More recently, *entropy-based* methods (Cui et al., 2025; Cheng et al., 2025) treat output entropy as a proxy for exploration. While these methods yield valuable insights, entropy remains a coarse and incomplete measure that cannot capture fine-grained exploration behavior.

We take an alternative perspective based on the mechanism of next-token prediction. At each decoding step, an LLM predicts a probability distribution over its entire vocabulary, offering a direct and fine-grained view of the its exploration behavior. How probability mass is distributed across vocabulary candidates, determines whether the model is exploring multiple reasoning paths or collapsing into a single deterministic trajectory. Unfortunately, capturing the full distribution is computationally prohibitive, as modern vocabularies often exceed 100K candidates. This practical constraint likely explains why prior work has favored scalar measures like entropy over the full distribution.

In this work, we revisit the token-level probability distribution. Our empirical evidence shows that these distribution are highly skewed, as illustrated in Figure 1, with only a few candidates carrying non-negligible probability mass. This finding justifies a tractable approximation: by focusing on the top-K candidates, we can effectively characterize exploration behavior. Building on this view, we further analyze the training dynamics of RLVR algorithms and uncover a salient trend: probability mass gradually concentrates on the top-1 candidate, while other candidates are suppressed. This over-concentration pushes the model toward deterministic behavior and, more importantly, directly explains the degradation of pass@K performance.

Motivated by this observation, we propose **Simple Pass@K Optimization (SimKO)**, a method designed to explicitly mitigate distribution collapse. The core idea is to redistribute gradient updates across the top-K candidates rather than allowing the top-1 to dominate. For verified-correct responses, SimKO shares positive gradients between the generated token and other high-probability candidates, reducing over-concentration on a single choice. For incorrect responses, SimKO applies stronger penalties to the top-1 candidate, encouraging probability mass to flow into alternative candidates. This asymmetric regularization proves especially effective when applied to "semantic forking" in the reasoning path, where token-level entropy is high.

We evaluate SimKO on several mathematical and logical reasoning benchmarks across multiple LLM backbones. SimKO consistently improves over vanilla GRPO and surpasses strong baselines across a wide range of K values (up to 256), as shown in Figure 1. Ablation studies further confirm the mechanism behind its gains, offering new evidence for the role of probability concentration in the exploration–exploitation trade-off. In summary, our work makes three key contributions:

- A new perspective on RLVR dynamics. We introduce top-K posterior probabilities as a tractable approximation of the full token-level distribution, providing direct insight into why existing RLVR methods often improve pass@1 at the cost of exploration.

- A simple yet principled method. We propose SimKO, which mitigates probability collapse through asymmetric gradient redistribution. By redistributing probability mass from overconfident top-1 tokens to other promising candidates, SimKO explicitly promotes exploration.

- Improved pass@K performance. We demonstrate SimKO's effectiveness on multiple math and logic benchmarks, showing consistent improvements in pass@K without sacrificing pass@1.

## 2 BACKGROUND AND PRELIMINARIES

Group Relative Policy Optimization (GRPO), a representative policy-based RLVR method (Shao et al., 2024), is a variant of PPO (Schulman et al., 2017) tailored for LLM fine-tuning. Given a question, the model generates $G$ different responses and updates its parameters according to the loss:

$$\mathcal{J}_{\text{GRPO}}(\theta; \gamma_{i,l}) = \frac{1}{G} \sum_{i=1}^{G} \frac{1}{|y_i|} \sum_{l=1}^{|y_i|} [\min\left(\gamma_{i,l} A_{i,l}, \text{clip}(\gamma_{i,l}, 1-\epsilon, 1+\epsilon) A_{i,l}\right) - \beta \mathbb{D}_{\text{KL}}(\pi_\theta || \pi_{\text{ref}})]$$

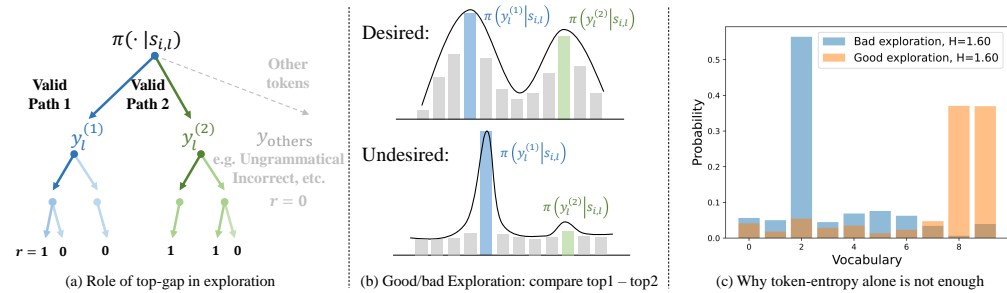

(a) Role of top-gap in exploration     (b) Good/bad Exploration: compare top1 – top2     (c) Why token-entropy alone is not enough

Figure 2: (a) The exploration behavior visualized according to the token-level posterior probability. (b) Comparison of two exploration strategy. (c) An example of two distributions with identical entropy but distinct probability distribution.

where $\gamma_{i,l} = \pi_\theta(y_{i,l} \mid s_{i,l}) / \pi_{\text{ref}}(y_{i,l} \mid s_{i,l})$ is the likelihood ratio between the current policy $\pi_\theta$ and the reference policy $\pi_{\text{ref}}$ at the $l$-th token of response $y_i$, with $s_{i,l} = (x, y_{i,<l})$ denoting the decoding state. The term $A_{i,l}$ represents the advantage of the $l$-th token in the $i$-th response, computed by normalizing rewards within the roll-out.

GRPO can be analyzed in gradient space under the learning-dynamics framework of Ren & Sutherland (2025). The dominant contribution to parameter updates comes from $\nabla_\theta A_{i,l} \gamma_{i,l}$. Here, the advantage $A_{i,l}$ can be viewed as a sequence-level adaptive learning rate that scales gradients from different responses $y_i$. Ignoring the contributions of the KL regularization and clipping terms (both primarily introduced for stability), the main optimization direction of most GRPO-based methods is governed by:

$$\nabla_\theta A_{i,l} \gamma_{i,l} = A_{i,l} \frac{\pi_\theta(y_{i,l}|s_{i,l})}{\pi_{\text{ref}}(y_{i,l}|s_{i,l})} \nabla_\theta \log \pi_\theta(y_{i,l}|s_{i,l}) = A_{i,l} \cdot \text{sg}(\gamma_{i,l}) \cdot \nabla_\theta \log \pi_\theta(y_{i,l}|s_{i,l}), \quad (1)$$

where $\text{sg}(\cdot)$ is the stopping-gradient operator.

## 3   OVER-CONCENTRATED DISTRIBUTION DEGRADES PASS@K SCORE

To understand the mechanism driving current RLVR methods to favor exploitation over exploration, we begin by examining the connection between the model's token-level probability distribution and its pass@K performance. We consider the illustrative example in Figure 2-(a), where the model generates the $l$-th token given the context $s_{i,l} = (x_i, y_{<l})$. Suppose two valid reasoning paths begin with $y_l^{(1)}$ and $y_l^{(2)}$, where $y_l^{(k)}$ denotes the rank-$k$ candidate under $\pi_\theta(\cdot \mid s_{i,l})$. A model with strong exploration ability will distribute probability more evenly between $\pi_\theta(y_l^{(1)} \mid s_{i,l})$ and $\pi_\theta(y_l^{(2)} \mid s_{i,l})$, as illustrated in the upper panel of Figure 2-(b).

To track the token probability distribution among top-K candidates, we propose to compute

$$\Lambda^{(k)} := \mathbb{E}_{x_i \sim \mathcal{D}, \, y \sim \pi_\theta(\cdot|s_{i,l})} \left[ \frac{1}{L} \sum_{l=1}^{L} \log \pi_\theta(y_l^{(k)} \mid s_{i,l}) \right], \qquad k \in \{1, \ldots, K\}. \quad (2)$$

This metric represents the average log-probability of the rank-$k$ candidate during generation. Despite the large vocabulary size, Figure 1 shows the probability mass is highly concentrated in the top few candidates. Empirically, we find that top-3 candidates (*i.e.* $k$=3) is a sufficient and efficient for approximating the distribution. Unlike entropy, which can take the same value when a single token dominates or when two tokens share similarly high probabilities (see Figure 2-(c)), this metric provides a more direct and interpretable view of how the model's posterior next-token distribution shifts. It therefore offers a clearer reflection of the model's probability landscape.

In addition to the probability of the rank-$k$ candidate, we track the evolution of the average log-probability of the token sequence that is actually sampled, which is denoted as $\Lambda$.[1] Since updates occur directly on these tokens, the gap between $\Lambda$ and other $\Lambda^{(k)}$ more precisely indicates exploitation: a larger gap reflects the model's tendency to concentrate on a single reasoning path.

---

[1] Just replace the $y_l^{(k)}$ term in Equation 2 by the sampled candidate.

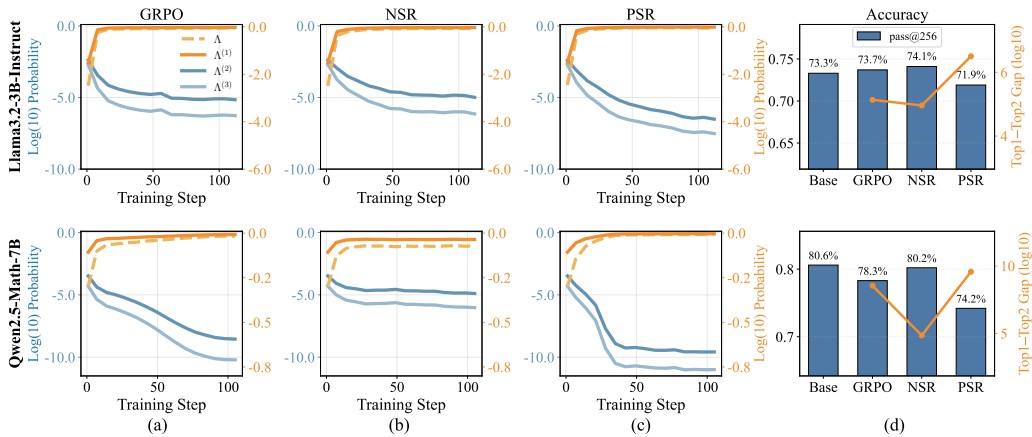

Figure 3: (a)-(c) Training dynamics of average log probability $\Lambda$ and top-K probabilities $\Lambda^{(k)}$ derived by GRPO, NSR, and PSR. (d) The corresponding pass@1 and pass@K results of the RLVL-trained models. Following the setups of Zhu et al. (2025), we train a Llama3.2-3B-Instruct on a mixture of GSM8K and MATH (Level 1) and train Qwen2.5-Math-7B on the MATH dataset.

To better disentangle the roles of positive and negative gradients, central to our algorithm design, we also conduct two ablations where the model is updated using only one type of response. We refer to these as Positive Sample Reinforce (PSR) and Negative Sample Reinforce (NSR) (Zhu et al., 2025).

We present the learning dynamics in Figure 3. The results in first column reveals a clear trend under GRPO training: the probability of the rank-1 candidate, $\Lambda^{(1)}$, saturates near 1, while the probabilities of alternatives ($\Lambda^{(2)}$, $\Lambda^{(3)}$) collapse to negligible values ($10^{-8}$–$10^{-10}$ in Qwen2.5-Math-7B). This demonstrates that GRPO aggressively concentrates probability mass on the single best candidate. While NSR partially mitigates this effect, PSR significantly exacerbates it.

Crucially, we observe a clear inverse relationship between the degree of probability concentration and pass@K performance. As shown in Figure 3-(d), as the probability gap between the rank-1 candidate and its alternatives increases, the model's pass@K accuracy declines. This suggests that over-concentration of posterior probability on the top-1 candidate suppresses the exploration of other plausible reasoning paths. This observation naturally motivates a key question: *If we can mitigate this over-concentration, can we improve pass@K performance by encouraging broader exploration?* Our algorithmic design is precisely driven by this intuition and presented in the next section.

## 4 SIMKO: SIMPLE PASS@K OPTIMIZATION

In this section, we present the three key components of our SimKO method: (i) identifying "forking" tokens for reasoning, (ii) applying top-K label smoothing to redistribute positive gradients among the top-K candidates, and (iii) strengthening rank-1 candidate updates for negative gradients. These steps, illustrated in Figure 4, work together to improve reasoning diversity.

### 4.1 IDENTIFYING INFORMATIVE TOKENS TO OPTIMIZE

While standard on-policy RLVR methods compute gradients over the entire generated sequence, not all tokens contribute equally to logical inference. Recent work (Wang et al., 2025) shows that only a small subset of tokens drives the core reasoning process, whereas the majority (over 80%) primarily ensure grammatical fluency and formatting. Our analysis supports this finding by showing that key reasoning tokens, often marked by "forking" points, often exhibit higher entropy and shape the reasoning trajectory (Figure 4-(a)). Greater diversity at these forking points naturally increases the likelihood that at least one of the K sampled trajectories is correct, thereby improving pass@K.

Building on this insight, SimKO selectively operate on a critical subset of tokens whose entropy is greater than a threshold $\tau$. In short, by replacing the $\gamma_{i,l}$ in the vanilla GRPO loss with a newly

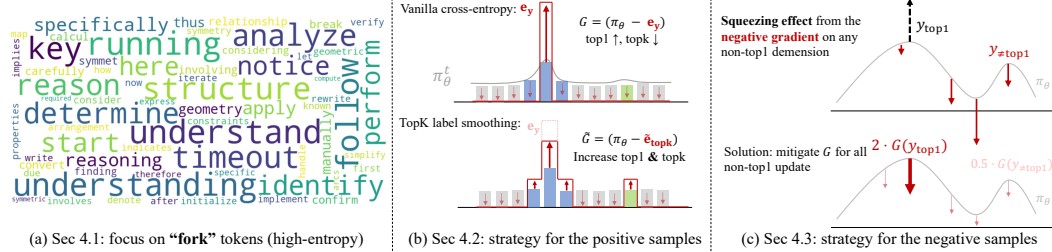

(a) Sec 4.1: focus on **"fork"** tokens (high-entropy)     (b) Sec 4.2: strategy for the positive samples     (c) Sec 4.3: strategy for the negative samples

Figure 4: Intuition of the proposed method. (a) We begin by identifying the "forking" tokens, which are high-entropy tokens, and diverge into multiple reasoning paths. (b) For positive samples, we redistribute the probability mass from the top-1 candidate to the top-K candidates, mitigating overconcentration. (c) For negative samples, we apply a strong penalty to the top-1 candidate and a weaker penalty to non-top-1 candidates to prevent the squeezing effect, thereby avoiding sharp distributions and facilitating model exploration.

defined $\tilde{\gamma}_{i,l}$, the proposed method can be written as:

$$\mathcal{J}_{\text{SimKO}}(\theta) = \mathcal{J}_{\text{GRPO}}(\theta; \gamma_{i,l} \to \tilde{\gamma}_{i,l}); \quad \tilde{\gamma}_{i,l} := \begin{cases} \gamma_{i,l}^{\text{pos}}, & \text{if } \mathcal{H}(\pi_\theta(\cdot|s_{i,l})) > \tau, \ A_{i,l} > 0, \\ \gamma_{i,l}^{\text{neg}}, & \text{if } \mathcal{H}(\pi_\theta(\cdot|s_{i,l})) > \tau, \ A_{i,l} < 0, \\ \gamma_{i,l}, & \text{if } \mathcal{H}(\pi_\theta(\cdot|s_{i,l})) \le \tau, \ \forall A_{i,l}, \end{cases} \quad (3)$$

where $\mathcal{H}(\pi_\theta(\cdot|s_{i,l})) = -\sum_{y\in\mathcal{V}} \pi_\theta(y \mid s_{i,l}) \log \pi_\theta(y \mid s_{i,l})$ denotes the entropy of the policy distribution at state $s_{i,l}$. $A_{i,l}$ is the advantage calculated in vanilla GRPO.

## 4.2 REDISTRIBUTING POSITIVE GRADIENTS AMONG TOP-K CANDIDATES

We now design regularization mechanisms for tokens with positive gradients, i.e., tokens from correct responses. To understand how the model's predictions evolve after an update, we analyze its behavior in gradient space and denote the derivative of the loss with respect to the logits $\mathbf{z}$ as $\mathcal{G}$-term:

$$\nabla_{\mathbf{z}} - \log \pi_\theta(y_{i,l}|s_{i,l}) := \mathcal{G}(i,l) = \pi_\theta(\cdot|s_{i,l}) - \mathbf{e}_{y_{i,l}}, \quad (4)$$

where $\mathbf{e}_{y_{i,l}}$ is the one-hot vector for the label $y_{i,l}$. We now analyze the one-step effect of learning the $l$-th token in the $i$-th correct response. As shown Figure 4-(b), one-step update increases the probability of the $y_{i,l}$-th candidate of $\pi_\theta(\cdot \mid s_{i,l})$ while simultaneously decreasing all other candidates.

When $y_{i,l}$ is the rank-1 candidate, which is highly likely under on-policy sampling, the distribution becomes sharper, and the probability gap between the rank-1 candidate and its alternatives grows. Continued training under this dynamic causes the rank-1 candidate to absorb nearly all probability mass, leaving the model unable to generate diverse yet correct responses that begin with rank-$k$ candidates. This effect is empirically validated by our results in Figure 3.

To address this issue, we need to design a mechanism that can reallocate probability mass absorbed by the rank-1 candidate back to other plausible ones, thereby restoring diversity. This resonates with the classical solution for over-concentration: label smoothing (Müller et al., 2019), in which the one-hot target $\mathbf{e}_y$ is replaced by a convex combination of the one-hot and a uniform distribution: $(1-\alpha)\mathbf{e}_y + \frac{\alpha}{V-1}\mathbf{u}$, where $\alpha \in [0,1]$ controls the smoothing strength.

However, directly applying vanilla label smoothing in LLM fine-tuning is problematic because the vocabulary size $V$ is extremely large. Spreading probability mass uniformly risks generating ungrammatical or irrelevant tokens, which can destabilize training. To address this problem, we take inspiration from the design of top-K sampling (Fan et al., 2018), and propose top-K label smoothing, which redistributes probability only across several most plausible candidates. Concretely, by replacing the $\mathbf{e}_y$ in Eqn. 4, we propose the $\tilde{\mathcal{G}}$-term:

$$\tilde{\mathcal{G}}(i,l) = \pi_\theta(\cdot|s_{i,l}) - \tilde{\mathbf{e}}_{\text{topk}} = \pi_\theta(\cdot|s_{i,l}) - ((1-\alpha)\mathbf{e}_y + \frac{\alpha}{K}\sum_{k\in\mathcal{I}_{\text{topk}}} \mathbf{e}_k), \quad (5)$$

where $\mathcal{I}_{\text{topk}}$ denotes the indices of the top-K tokens under the current model distribution. Importantly, we retain this definition even if $y \in \mathcal{I}_{\text{topk}}$, ensuring consistent treatment of the target token. We now

present a equivalent loss function with the re-designed one-step gradient provided in Equation 5 as

$$\gamma_{i,l}^{\text{pos}} = (1-\alpha)\,\gamma_{i,t} \;+\; \frac{\alpha}{|\mathcal{I}_{\text{topk}}|} \sum_{k \in \mathcal{I}_{\text{topk}}} \text{sg}\!\left(\frac{\gamma_{i,l}}{\gamma_{i,l}^{(k)}}\right) \gamma_{i,l}^{(k)}, \quad \alpha \in [0,1], \tag{6}$$

where $\gamma_{i,l}^{(k)} = \pi_\theta\big(y_{i,l}^{(k)} \mid s_{i,t}\big)/\pi_{\theta_{\text{ref}}}\big(y_{i,l}^{(k)} \mid s_{i,l}\big)$, which is the ratio of the model's probability on the rank-$k$ candidate. The term $\text{sg}(\cdot)$ is designed to make sure that $\text{sg}(\gamma_{i,l}^{\text{pos}}) = \text{sg}(\gamma_{i,l})$, which is crucial for importance sampling in GRPO-based methods (see Equation 1). More design details can be found in Appendix B. Intuitively, optimizing with $\gamma_{i,l}^{\text{pos}}$ increases the probabilities of the top-K candidates, causing the output distribution to form a plateau rather than a sharp peak. This flatter distribution promotes exploration and enhances response diversity, as illustrated in Figure 4.

### 4.3 RESTRAINING NEGATIVE GRADIENTS ON TOP-1 CANDIDATE

We design a separate mechanism for negative responses because their gradients affect the probability distribution asymmetrically compared to positive gradients. As in the previous subsection, we analyze the one-step influence of negative gradients ($A_{i,l} < 0$) through their $\mathcal{G}$-term. Expanding $\mathcal{G}(i,l)$ element-wise (Equation 4) leads to: $[\mathcal{G}(i,l)]_{y_{i,l}} = \pi_\theta(y|s_{i,l}) - 1$ and $[\mathcal{G}(i,l)]_{\text{others}} = \pi_\theta(y|s_{i,l})$. This formulation highlights two key effects. First, candidates with already high probabilities experience minimal push-down pressure when selected, since $\pi_\theta(y_{i,l} \mid s_{i,l}) - 1 \approx 0$. Second, the probability mass removed from the target candidate is redistributed proportionally across all other candidates based on their $\pi_\theta(y \mid s_{i,l})$. These mechanisms align with the "squeezing effect" described in Ren & Sutherland (2025), where the rank-1 candidate benefits the most during redistribution.

Prior approaches apply a uniform scaling to all negative gradients (Shao et al., 2024; Yu et al., 2025). In particular, NSR (Zhu et al., 2025) adopts the strategy of simply amplifying negative gradients, e.g., by multiplying a big scalar $c$ to those $A_{i,l} < 0$, with the goal of enhancing exploration and mitigating over-concentration. However, as our analysis in negative gradient shows, this strategy is ineffective in mitigating the decay of exploration. While uniformly stronger push-down pressure can suppress probability growth of the rank-1 candidate, the induced squeezing effect on non-rank-1 candidates paradoxically makes the distribution sharper, as demonstrated in the upper panel of Figure 4-(c). In summary, for negative gradients, we must control the relative strength between the negative gradients imposed on rank-1 and non-rank-1 candidates.

Motivated by this, we propose to replace the original $\gamma_{i,l}$ by $\lambda \cdot \gamma_{i,l}$ when $y_{i,l}$ is the rank-1 candidate, where $\lambda$ is a hyper-parameter that is greater than one. The effect of applying this $\gamma_{i,l}^{\text{neg}}$ is demonstrated by the lower panel in Figure 4-(c). This targeted adjustment forms a key conceptual difference from prior methods relying on uniform negative-gradient amplification.

## 5 EXPERIMENTS AND RESULTS

### 5.1 EXPERIMENTAL SETUP

**Models and Datasets.** We experiment with a diverse set of models, including Qwen2.5-7B (Team, 2024), Qwen2.5-Math-7B (Yang et al., 2024), and Llama3.2-3B-Instruct (AI@Meta, 2024). Following the setups in Zhu et al. (2025); Zeng et al. (2025), the Qwen models are trained on the MATH dataset (Hendrycks et al., 2021), while the Llama model is trained on a combined dataset of GSM8K (Cobbe et al., 2021) and MATH (Level 1). For logical reasoning tasks, we train Qwen2.5-7B on the Synlogic-easy dataset (training split).

**Training Details.** All models are trained with a learning rate of $10^{-6}$, a batch size of 1024, and a PPO mini-batch size of 256. Each input problem is sampled with 8 responses using temperature 1.0. We set $\alpha$ in SimKO as 0.01, and define the entropy threshold $\tau(q)$ as the $q$-quantile of the token-level entropy distribution, such that a fraction $q$ of all tokens have entropy values lower than $\tau(q)$. Unless otherwise specified, we use $\tau(0.8)$ in our experiments. We also set $\lambda_{\text{top1}} = 1.1$, except for Qwen2.5-7B where we use $\lambda_{\text{top1}} = 1.05$. For the logic task, we apply a warm-up of 50 steps and use a smaller $\alpha = 0.005$ along with $\lambda_{\text{top1}} = 1.05$.

**Evaluation Protocol.** We compare SimKO against several competitive baselines, including GRPO, PSR, NSR, W-REINFORCE (Zhu et al., 2025), KL-Cov (Cui et al., 2025), P@k T. (Chen et al.,

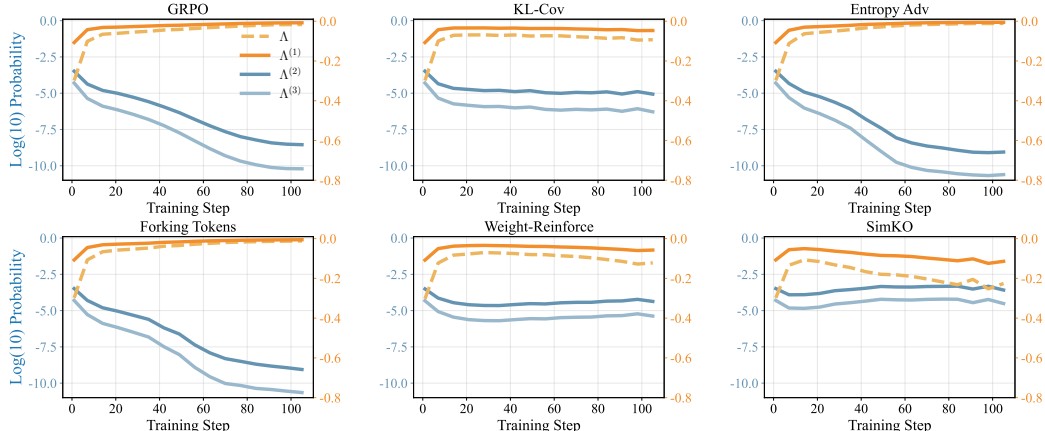

Figure 5: Comparison of SimKO with GRPO, KL-Cov, "forking" tokens, Weight-Reinforce and Entropy-Adv on Qwen2.5-Math-7B. SimKO effectively controls probability concentration on the $\Lambda^{(1)}$ while preserving diversity among $\Lambda^{(2)}$ and $\Lambda^{(3)}$.

2025a), with "forking" tokens (Wang et al., 2025) and Entropy-Adv (Cheng et al., 2025). Evaluations are conducted on a variety of reasoning benchmarks: MATH-500 (Hendrycks et al., 2021), Minerva Math (Lewkowycz et al., 2022), Olympiad-Bench (He et al., 2024), AMC, AIME, Synlogic-easy (validation split) (Liu et al., 2025a), and BBH (Suzgun et al., 2022). To obtain comprehensive evaluation, we adopt the unbiased pass@K metric with K up to 256, computed as pass@K $:= \mathbb{E}_{x \sim \mathcal{D}} \left[ 1 - \binom{n-c}{K} / \binom{n}{K} \right]$, where $c$ denotes the number of correct completions out of $n$ generated responses. To reduce evaluation variance on small datasets (e.g., AIME and AMC), we set $n = 300$; for other math datasets, we use $n = 256$, and for logic datasets, $n = 128$.

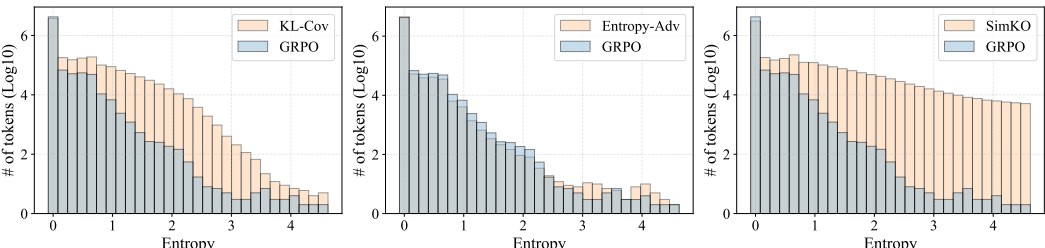

Figure 6: Token-level entropy distributions from the Qwen2.5-Math-7B backbone trained with SimKO, GRPO, KL-Cov and Entropy-Adv, demonstrating SimKO's ability to maintain the entropy of the "forking" tokens.

## 5.2 EFFECTS OF SIMKO ON TRAINING DYNAMICS

We analyze the training dynamics of SimKO in comparison with GRPO, KL-Cov, "forking" tokens, Weight-Reinforce and Entropy-Adv. Figure 5 presents the changes of top-K log-probabilities ($\Lambda^{(1)}$, $\Lambda^{(2)}$, and $\Lambda^{(3)}$) across training steps.

As can be seen, GRPO leads to severe over-concentration: $\Lambda^{(1)}_{\text{GRPO}}$ increases to nearly 1, while $\Lambda^{(2)}_{\text{GRPO}}$ and $\Lambda^{(3)}_{\text{GRPO}}$ sharply drop below $10^{-8}$ and $10^{-10}$, respectively. This indicates that nearly all probability mass collapses onto the top-1 token. KL-Cov exhibits a moderate concentration effect due to the KL penalty, while Entropy-Adv collapses even more rapidly, likely because of its stronger emphasis on high-entropy tokens. Training only on forking tokens further exacerbates probability over-concentration compared to GRPO, as it selectively increases the probability of high-entropy tokens. Weight-Reinforce alleviates part of the over-concentration, but its sequence-level penalization of negative samples induces the squeezing effect discussed in Section 4.3, leaving it still prone to probability concentration. In contrast, SimKO achieves the most effective deconcentration among all methods. This is evidenced by a lower $\Lambda^{(1)}_{\text{SimKO}}$ and higher $\Lambda^{(2)}_{\text{SimKO}}$ and $\Lambda^{(3)}_{\text{SimKO}}$. These results suggest that SimKO effectively mitigates probability mass collapse and can potentially encourages exploration during training.

| Method | 1 | 2 | 4 | 8 | 16 | 32 | 64 | 128 | 256 |
|---|---|---|---|---|---|---|---|---|---|
| *Qwen2.5-Math-7B* | | | | | | | | | |
| Base Model | 25.8 | 35.9 | 45.1 | 52.8 | 59.1 | 64.3 | 68.8 | 72.7 | 76.4 |
| GRPO | 41.7 | 47.9 | 53.3 | 58.1 | 62.6 | 66.4 | 69.7 | 72.9 | 76.1 |
| PSR | 39.5 | 45.1 | 49.9 | 54.3 | 58.5 | 62.4 | 66.0 | 69.3 | 72.5 |
| NSR | 39.5 | 47.0 | 53.6 | 59.2 | 64.0 | 68.2 | 72.2 | 76.1 | 80.3 |
| W-REINFORCE | 41.5 | 48.6 | 54.4 | 59.5 | 64.1 | 68.4 | 72.4 | 76.3 | 80.2 |
| KL-Cov | 42.5 | 49.5 | 55.4 | 60.4 | 64.8 | 68.6 | 72.0 | 75.5 | 79.0 |
| P@k T. | 39.8 | 47.7 | 54.1 | 59.5 | 64.2 | 68.5 | 72.5 | 76.3 | 80.1 |
| GRPO w/forking tokens | 41.8 | 47.6 | 52.8 | 57.3 | 61.4 | 65.2 | 68.7 | 72.4 | 76.8 |
| Entropy-Adv | 42.1 | 47.7 | 52.6 | 56.7 | 60.3 | 63.9 | 67.7 | 71.9 | 76.1 |
| SimKO | **43.4** | **50.7** | **56.7** | **61.4** | **65.6** | **69.4** | **73.1** | **76.7** | **80.5** |
| Δ(SimKO-GRPO) | +1.7 | +2.8 | +3.4 | +3.3 | +3.0 | +3.0 | +3.4 | +3.8 | +4.4 |
| *Qwen2.5-7B* | | | | | | | | | |
| Base Model | 26.6 | 35.0 | 42.7 | 49.5 | 55.6 | 61.1 | 66.2 | **71.3** | **76.4** |
| GRPO | 38.4 | 44.4 | 49.8 | 54.5 | 58.7 | 62.5 | 66.0 | 69.3 | 72.3 |
| PSR | 36.2 | 41.6 | 46.5 | 51.2 | 55.7 | 59.8 | 63.4 | 66.9 | 70.7 |
| NSR | 35.2 | 42.1 | 48.2 | 53.7 | 58.4 | 62.5 | 66.3 | 69.8 | 72.8 |
| W-REINFORCE | 35.9 | 42.7 | 48.8 | 54.0 | 58.8 | **63.3** | **67.3** | 71.1 | 75.2 |
| SimKO | **38.9** | **45.5** | **50.8** | **55.2** | **59.2** | 63.1 | 67.0 | 70.8 | 74.3 |
| Δ(SimKO-GRPO) | +0.5 | +1.1 | +1.0 | +0.7 | +0.5 | +0.6 | +1.0 | +1.5 | +2.0 |
| *Llama3.2-3B-Instruct* | | | | | | | | | |
| Base Model | 14.2 | 20.7 | 28.0 | 35.6 | 43.1 | 50.2 | 57.2 | 63.6 | 68.9 |
| GRPO | 23.3 | 29.4 | 35.7 | 41.4 | 46.9 | 52.4 | 57.9 | 63.7 | 69.5 |
| PSR | 20.6 | 26.1 | 32.0 | 37.9 | 43.9 | 49.8 | 55.8 | 61.8 | 67.8 |
| NSR | 22.5 | 29.1 | 35.6 | 41.6 | 47.4 | 53.1 | 58.6 | 64.1 | 69.7 |
| W-REINFORCE | 22.4 | 28.8 | 34.9 | 40.8 | 46.4 | 52.0 | 57.5 | 63.1 | 68.1 |
| SimKO | **24.0** | **30.3** | **36.4** | **42.0** | **47.6** | **53.3** | **58.9** | **64.8** | **70.8** |
| Δ(SimKO-GRPO) | +0.7 | +0.9 | +0.7 | +0.6 | +0.7 | +0.9 | +1.0 | +1.2 | +1.3 |

Table 1: Average pass@256 results for Qwen2.5-Math-7B, Qwen2.5-7B, and Llama3.2-3B-Instruct on MATH500, AIME 2024/25, Minerva_math, Olympiadbench, and AMC23 Datasets.

To further validate this, we visualize the histogram of token-level entropy in Figure 6. GRPO drives most tokens toward near-zero entropy. SimKO, however, can preserve token entropy, particularly at "semantic forks", where high entropy is desirable for exploration. This preservation of entropy further further confirms SimKO's role in promoting exploration.

## 5.3 MAIN RESULTS ON MATH BENCHMARKS

We evaluate SimKO on six widely used math benchmarks across different model backbones. Table 1 reports average pass@K results for $K$ ranging from 1 to 256. Detailed results on separate benchmarks are provided in the Appendix Table 4.

Compared to the base models, SimKO significantly improves the pass@1 score by 17.6% on Qwen2.5-Math-7B and 9.8% on Llama3.2-3B-Instruct, indicating improved exploitation. At the same time, it also boosts the pass@256 score by 4.1% and 1.9% on the same backbones respectively, demonstrating improved exploration and overall reasoning quality. Although the base model of Qwen2.5-7B achieves the highest pass@256 score (76.4% vs. 74.3% from SimKO), its pass@1 performance is notably low (26.6% vs. 38.9% from SimKO), indicating an imbalance between exploration and exploitation.

Compared to GRPO and its variants (KL-Cov, Entropy-Adv, with "forking" tokens and P@k T.), SimKO consistently outperforms them across all model backbones and values of K. More importantly, SimKO delivers these gains without sacrificing exploration (pass@256) and with even stronger exploitation (pass@1). Relative to GRPO, SimKO improves pass@256 by 4.4%, 2.0%, and 1.3% on Qwen2.5-Math-7B, Qwen2.5-7B, and Llama3.2-3B-Instruct, respectively, while also

| Method | Synlogic | | | | | | | | BBH | | | | | | | |
|---|---|---|---|---|---|---|---|---|---|---|---|---|---|---|---|---|
| | 1 | 2 | 4 | 8 | 16 | 32 | 64 | 128 | 1 | 2 | 4 | 8 | 16 | 32 | 64 | 128 |
| Base Model | 3.1 | 4.9 | 7.5 | 11.1 | 15.4 | 20.0 | 24.5 | 28.7 | 42.4 | 59.3 | 74.4 | **84.6** | **89.9** | **92.4** | **93.6** | **94.2** |
| GRPO | **35.3** | _38.2_ | _40.8_ | _42.9_ | _44.7_ | _46.3_ | _47.9_ | _49.4_ | _56.4_ | _64.6_ | _71.3_ | 76.6 | 80.7 | 83.9 | 86.3 | 88.2 |
| PSR | 27.3 | 29.3 | 31.7 | 34.3 | 36.9 | 39.4 | 41.6 | 43.6 | 54.9 | 62.7 | 68.8 | 73.4 | 76.8 | 79.4 | 81.4 | 82.8 |
| NSR | 1.1 | 1.9 | 3.2 | 5.3 | 8.5 | 12.5 | 17.2 | 21.6 | 26.1 | 41.6 | 59.3 | 74.9 | 85.3 | _90.5_ | _92.7_ | _93.5_ |
| W-REINFORCE | 0.8 | 1.3 | 1.8 | 2.4 | 2.9 | 3.4 | 3.8 | 3.9 | 15.4 | 21.9 | 27.8 | 32.3 | 35.4 | 37.2 | 38.3 | 38.8 |
| SimKO | _34.7_ | **38.4** | **42.0** | **45.5** | **48.5** | **51.0** | **53.2** | **55.0** | **58.4** | **69.5** | **77.7** | _83.2_ | _86.8_ | 89.2 | 90.9 | 92.0 |

Table 2: Pass@K results for Qwen2.5-7B on Synlogic and BBH Datasets.

achieving higher pass@1. Training solely with "forking" tokens does not improve pass@K and can even underperform GRPO. Although P@k T. achieves a higher pass@256 (80.1%), its pass@1 performance drops to 39.8%, indicating that it fails to balance exploration and exploitation effectively.

For NSR and W-REINFORCE, strong pass@256 performance is maintained, but often at the expense of much lower pass@1. In contrast, SimKO achieves a better balance on most backbones. On Qwen2.5-Math-7B, SimKO reaches a slightly higher pass@256 score (80.5% vs. 80.3% for NSR and 80.2% for W-REINFORCE) while clearly outperforming both in pass@1 (43.4% vs. 39.5% and 41.5%). A similar trend is observed on Llama3.2-3B-Instruct, where SimKO improves both pass@256 (70.8% vs. 69.7% and 68.1%) and pass@1 (24.0% vs. 22.5% and 22.4%). For Qwen2.5-7B, however, the trade-offs differ: SimKO outperforms NSR on pass@256 (74.3% vs. 72.8%) but lags slightly behind W-REINFORCE (75.2%), and on pass@1 it significantly surpasses both baselines (35.2% vs. 35.9% for NSR and 38.9% for W-REINFORCE).

These results support our hypothesis that alleviating probability over-concentration (Figure 5) improves pass@K performance, indicating a better balance between exploitation and exploration.

## 5.4 GENERALIZATION TO LOGICAL TASKS

We evaluate SimKO's generalization ability on two logic reasoning benchmarks, as shown in Table 2. These benchmarks cover two scenarios: (1) Synlogic, an in-distribution task where the base model performs poorly in pass@K on both the training and test datasets, which come from the same distribution (Synlogic-easy), and (2) BBH, an out-of-distribution task where the base model performs better in pass@K, but the test data differs from the training data distribution.

On Synlogic, SimKO significantly outperforms the base model, with a +31.6% gain in pass@1 and +26.3% at pass@128. Methods like GRPO and PSR show improvements but lag behind SimKO by 4.6% and 11.4% at pass@128. NSR and W-REINFORCE, however, fail to train effectively, with pass@1 scores of only 1.1% and 0.8%. Similar observations can also be found in BBH dataset.

On BBH, SimKO boosts the base model's pass@1 to 58.4% (+16.0%), and maintains stability at higher sampling rates, with just a 2.2% decrease in pass@128. GRPO and PSR, by comparison, drop 6.0% and 11.4% at pass@128 compare to base model, showing difficulties in sustaining performance. NSR and W-REINFORCE perform poorly, achieving only 26.1% and 15.4% at pass@1.

These results demonstrate that relying solely on negative samples is insufficient to improve Pass@K on challenging tasks. In contrast, SimKO exhibits strong generalization, effectively trains on difficult tasks, and improves Pass@K performance by mitigating probability over- concentration.

## 5.5 ABLATION STUDIES

We conduct an in-depth analysis of how various parameters, such as $\tau$, $\alpha$, and the impacts of $\gamma^{\text{pos}}$ and $\gamma^{\text{neg}}$, affect SimKO. The full ablation results are summarized in Table 3, and the performance variations with respect to $\tau$ and $\alpha$ are shown in Figure 7. Specifically, we evaluate $\alpha$ values from 0 to 0.1, where $\alpha = 0$ represents the performance of GRPO. The results show that as $\alpha$ increases, pass@256 consistently improves over GRPO, while pass@1 experiences a slight reduction. Notably, the performance remains relatively stable within this range of $\alpha$, demonstrating the robustness of our method against moderate changes in $\alpha$. For $\tau$, we test values from $\tau(0)$ to $\tau(1)$, with $\tau(1)$ corresponding to GRPO. Notably, SimKO outperforms GRPO across all $\tau$ values in pass@256. However,

| Method | AIME24 | AIME25 | AMC | MATH500 | Minerva | Olympiad | Avg. |
|--------|--------|--------|-----|---------|---------|----------|------|
| Base Model | 13.2/66.0 | 5.4/51.8 | 38.2/98.5 | 55.8/96.0 | 16.5/68.8 | 25.6/77.0 | 25.8/76.4 |
| GRPO | 28.1/72.3 | 11.5/52.1 | 61.2/97.1 | 76.6/96.2 | 33.4/64.0 | 39.1/74.7 | 41.7/76.1 |
| $\alpha$=0.02 | 31.1/79.9 | **13.3**/58.9 | 62.6/99.3 | **77.6**/**97.0** | 35.2/66.2 | 39.4/76.9 | 43.2/79.7 |
| $\alpha$=0.03 | 30.9/72.7 | 12.4/64.6 | 62.4/97.5 | 77.2/97.6 | 34.9/66.9 | 38.9/76.9 | 42.8/79.4 |
| $\alpha$=0.05 | 30.8/78.7 | 12.2/**67.8** | 63.0/97.5 | 77.2/96.8 | 34.8/65.1 | 39.3/76.3 | 42.9/80.4 |
| $\alpha$=0.1 | 29.1/75.5 | 12.4/65.0 | 61.8/**99.9** | 77.2/96.8 | 35.2/67.6 | 38.9/**77.8** | 42.4/80.4 |
| $\tau(0)$ | 10.7/74.1 | 6.2/54.0 | 51.7/96.7 | 70.5/95.8 | 32.2/67.3 | 33.5/74.5 | 34.1/77.1 |
| $\tau(0.4)$ | 20.8/70.9 | 7.2/57.5 | 58.3/94.6 | 74.9/95.2 | 33.8/68.0 | 36.4/75.0 | 38.6/76.9 |
| $\tau(0.6)$ | 27.5/77.6 | 10.0/56.0 | 62.2/**99.9** | 76.5/96.8 | **35.3**/**68.8** | 38.2/76.4 | 41.6/79.3 |
| w/o $\gamma^{\text{neg}}$ | 31.5/**80.7** | 11.5/57.9 | **62.7**/97.4 | 77.1/96.2 | 34.1/65.8 | 39.0/75.6 | 42.8/78.9 |
| w/o $\gamma^{\text{pos}}$ | 30.4/75.2 | 12.7/64.9 | 62.3/99.6 | 77.4/96.4 | 34.7/65.8 | 39.5/77.3 | 42.8/79.9 |
| SimKO | **32.8**/78.0 | 12.9/64.6 | 62.4/97.5 | **77.6**/96.8 | 35.0/68.4 | **39.8**/**77.8** | **43.4**/**80.5** |

Table 3: Ablations on $\alpha$, $\tau$, $\gamma^{\text{pos}}$, and $\gamma^{\text{neg}}$. Pass@1/Pass@256 scores are evaluated using Qwen2.5-Math-7B.

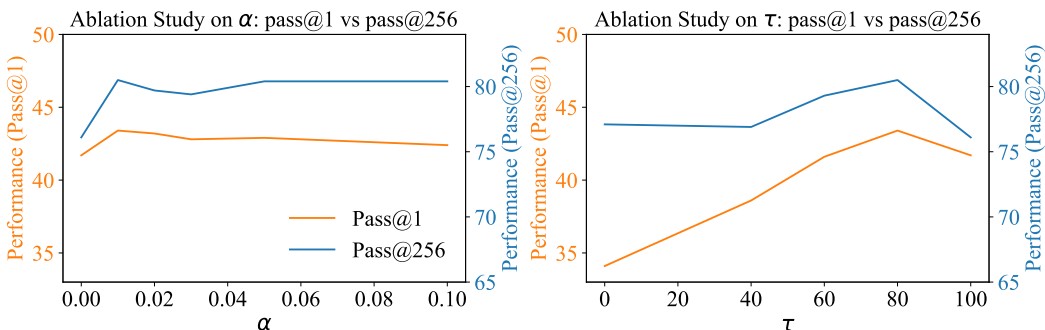

Figure 7: Ablations on $\alpha$ and $\tau$. Pass@1 and pass@256 scores are evaluated using the Qwen2.5-Math-7B backbone on math benchmarks.

when $\tau(0)$, where SimKO is applied to all tokens, pass@1 drops significantly. This indicates that restricting SimKO to the "semantic forks" of the task is essential for maintaining optimal performance. Additionally, as shown in Table 3, applying SimKO exclusively to either correct or incorrect examples leads to a drop in pass@K performance. This highlights the importance of asymmetric regularization, applied to both correct and incorrect examples, as it yields the best results.

## 6  CONCLUDING REMARKS

This paper addresses a key limitation in current RLVR methods, where pass@K performance drops due to reduced output diversity. Through analyzing token-level posterior probabilities, we find that RLVR training causes the model to overly concentrate probability mass on the top-1 candidate, leading to a deterministic policy and limited exploration. To overcome this issue, we propose Simple Pass@K Optimization (SimKO), a method that mitigates this effect by redistributing gradient updates across the top-K candidates. Extensive evaluations demonstrate that SimKO effectively preserves output diversity and consistently outperforms the GRPO baseline on both pass@1 and pass@256 metrics. These results highlight that SimKO achieves a superior balance between exploitation and exploration, thereby enhancing the model's overall reasoning capabilities.

ICLR PAPER CHECKLIST

1. **Ethics Statement**

   **Answer:** This research fully adheres to the ICLR Code of Ethics. The study does not involve human subjects or the use of personal or sensitive data. All datasets and code utilized and released conform to their respective licenses and terms of use. The contributions in this work are foundational and do not raise issues related to fairness, privacy, security, or potential misuse. We confirm that all ethical considerations have been thoroughly addressed.

2. **Reproducibility Statement**

   **Answer:** We are committed to making our work easy to reproduce. All necessary details for replicating our main experimental results— including data access, experimental setup, model configurations, and evaluation metrics— will be uploaded to GitHub after the paper is accepted. Users can follow our documentation and scripts to accurately reproduce the results, ensuring transparency and scientific rigor.

3. **The Use of Large Language Models**

   **Answer:** We utilize large language models to assist in enhancing our writing by checking for typos, improving the fluency of expressions, and ensuring that the language adheres to academic standards. This helps make the writing more structured and professional, ultimately making it easier for the audience to understand the content.

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

## A  RELATED WORKS

### A.1  REINFORCEMENT LEARNING WITH VERIFIABLE REWARDS IN LLMS

Reinforcement learning with verifiable rewards (RLVR) from large language models (LLMs) has demonstrated significant potential (DeepSeek-AI, 2025; Hugging Face, 2025; Zeng et al., 2025; He et al., 2025), especially when directly applied to a base model using GRPO (Shao et al., 2024) for RL training. This approach has notably enhanced the base model's performance, particularly in improving its reasoning abilities for mathematical and coding tasks. Subsequent works have focused on improving GRPO to further enhance the algorithm's performance. For instance, DAPO (Yu et al., 2025) independently adjusts GRPO's clipping thresholds and removes KL regularization to encourage larger updates in correct answer. Dr.GRPO (Liu et al., 2025b) eliminates the normalization term when computing advantages to prevent length bias. GSPO (Zheng et al., 2025) modifies the importance sampling from the token level to the sequence level, which proves to be more stable in training mixture-of-experts (MoE) models. These modifications have contributed to improvements in the model's pass@1 performance, but they have not specifically addressed pass@K performance, which relates to the model's exploration ability.

### A.2  EFFECTIVE EXPLORATION FOR RLVR IN LLMS

A central challenge in RLVR tasks lies in moving beyond the exploitation of a pretrained model's implicit knowledge to actively exploring diverse reasoning paths. Current methods tend to converge on a limited set of solutions, as evidenced by poor performance on the pass@K metric, which evaluates the coverage of multiple reasoning paths and thus reflects exploration effectiveness (Yue et al., 2025; Wu et al., 2025). To address this exploration deficit, the community has pursued several strategies. *Data-centric* methods aim to use data augmentation to enhance the model's exposure to diverse reasoning environments, thereby encouraging the exploration of a broader range of solution paths. One such approach involves using off-policy data from more capable models to expand the model's knowledge and promote solution diversity (Dong et al., 2025; Li et al., 2025). Additional strategies include generating varied responses for challenging samples (Yang et al., 2025) or paraphrasing questions to stimulate different reasoning trajectories for the same problem (Liang et al., 2025). As a complementary approach, *Reward-centric* methods redesign the objective function to directly incentivize diversity by calculating a group-wise reward based on a set of candidate solutions, providing an unbiased gradient for optimizing Pass@K (Walder & Karkhanis, 2025; Chen et al., 2025b). While these methods are effective to some extent, both treat the model as a black box, manipulating its inputs and final supervisory signals without understanding the internal mechanisms driving exploration. To address this limitation, a more recent line of work has shifted focus inward. The *Entropy-based* methods using entropy as a proxy for behavioral diversity and exploration (Cui et al., 2025; Cheng et al., 2025; Wang et al., 2025; Hou et al., 2025; Hu et al., 2025). However, policy entropy is a rough and indirect measure that does not allow for fine-grained observation of changes in the model's distribution. This limitation highlights the need for a mechanism that can directly monitor and explore how the distribution of next-token predictions changes during the model's training process, which is the central focus of our work.

Another relevant line of work decomposes RLVR updates into positive and negative components, arguing that positive reinforcement sharpens the distribution while negative reinforcement enhances exploration (Zhu et al., 2025). However, as we analyze in Section 4.3, negative gradients do not uniformly promote exploration: when applied to low-probability tokens, they can in fact sharpen the distribution and induce a squeezing effect, which was not captured in prior analyses. Moreover, as shown in Sections 5.3 and 5.4, negative-only reinforcement improves exploration but at the cost of exploitation. It increases pass@K but consistently decreases pass@1. In particular, on tasks where the model must first consolidate exploitation before exploration becomes beneficial, strong negative reinforcement disrupts early pattern learning, ultimately reducing both pass@1 and pass@K. Our work addresses these limitations by providing a token-level, learning-dynamics–based perspective that explains these behaviors and motivates our proposed approach.

# B MORE ABOUT THE DESIGN OF $\gamma_{i,l}^{\text{POS}}$

**Derivation of Equation 4**: To calculate the gradient to the logits, i.e., $\nabla_{\mathbf{z}} \log \pi_\theta(y_{i,l}|s_{i,l})$, we first need to recall how the probability $\pi$ is calculated. In most LLMs, the last layer is usually a Softmax function, which can convert a length-$v$ logits vector $\mathbf{z}$ to a categorical distribution vector $\pi$ with the same length. Then,

$$\nabla_{\mathbf{z}} \log \pi_\theta(y_{i,l}|s_{i,l}) = \nabla_{\mathbf{z}} \log \frac{\mathbf{e}^{z_{y_{i,l}}}}{\sum_i \mathbf{e}^{z_i}} = \nabla_{\mathbf{z}} \left( z_{y_{i,l}} - \log \sum_k \mathbf{e}^{z_k} \right).$$

We can then calculate the dimension for $k = y_{i,l}$ and $k \neq y_{i,l}$ separately. Specifically, for $k = y_{i,l}$, we have $\frac{\partial \log \pi_\theta(y_{i,l}|s_{i,l})}{\partial z_k} = 1 - \frac{\partial \log \sum_k e^{z_k}}{\partial z_k} = 1 - \pi_k$. For $k \neq y_{i,l}$, we only have $\frac{\partial \log \pi_\theta(y_{i,l}|s_{i,l})}{\partial z_k} = 0 - \pi_k$. Putting everything together and considering the negative sign in the definition, we have $\nabla_{\mathbf{z}} - \log \pi_\theta(y_{i,l}|s_{i,l}) := \mathcal{G}(i,l) = \pi_\theta(\cdot|s_{i,l}) - \mathbf{e}_{y_{i,l}}$, i.e., our Equation 4.

This appendix provides more details about how we design Equation-(6):

$$\gamma_{i,l}^{\text{pos}} = (1 - \alpha)\gamma_{i,t} + \frac{\alpha}{|\mathcal{I}_{\text{topk}}|} \sum_{k \in \mathcal{I}_{\text{topk}}} \text{sg}\left( \frac{\gamma_{i,l}}{\gamma_{i,l}^{(k)}} \right) \gamma_{i,l}^{(k)}, \quad \alpha \in [0,1],$$

Based on the smoothed $\tilde{\mathcal{G}}(i,l)$ term in Equation-(5):

$$\tilde{\mathcal{G}}(i,l) = \pi_\theta(\cdot|s_{i,l}) - \tilde{\mathbf{e}}_{\text{topk}} = \pi_\theta(\cdot|s_{i,l}) - \left( (1-\alpha)\mathbf{e}_y + \frac{\alpha}{K} \sum_{k \in \mathcal{I}_{\text{topk}}} \mathbf{e}_k \right),$$

Following the $\mathcal{AKG}$ decomposition in Ren & Sutherland (2025), we know

$$\nabla_\theta \gamma_{i,l} = \underbrace{A_{i,l} \cdot \text{sg}(\gamma_{i,l})}_{\text{Constant}} \cdot \underbrace{\nabla_{\mathbf{z}} \log \pi_\theta(y_i \mid s_{i,l})}_{\text{Defined as } \mathcal{G}(i,l)} = A_{i,l} \cdot \text{sg}(\gamma_{i,l}) \cdot \nabla_\theta \mathbf{z} \mathcal{G}(i,l).$$

Now, we replace $\mathcal{G}(i,l)$ term to $\tilde{\mathcal{G}}(i,l)$, the gradient part ($A_{i,l} \cdot \text{sg}(\gamma_{i,l})$ is a constant w.r.t $\theta$):

$$\nabla_\theta \mathbf{z} \tilde{\mathcal{G}}(i,l) = \nabla_\theta \mathbf{z} \left( \pi_\theta(\cdot|s_{i,l}) - \left( (1-\alpha)\mathbf{e}_y + \frac{\alpha}{K} \sum_{k \in \mathcal{I}_{\text{topk}}} \mathbf{e}_k \right) \right)$$

$$= \nabla_\theta \mathbf{z} \left( (1-\alpha)\pi_\theta(\cdot|s_{i,l}) + \alpha\pi_\theta(\cdot|s_{i,l}) - \left( (1-\alpha)\mathbf{e}_y + \frac{\alpha}{K} \sum_{k \in \mathcal{I}_{\text{topk}}} \mathbf{e}_k \right) \right)$$

$$= \nabla_\theta \mathbf{z} \left( (1-\alpha)(\pi_\theta(\cdot|s_{i,l}) - \mathbf{e}_y) + \frac{\alpha}{K} \sum_{k \in \mathcal{I}_{\text{topk}}} (\pi_\theta(\cdot|s_{i,l}) - \mathbf{e}_k) \right)$$

$$= (1-\alpha)\nabla_\theta \mathbf{z} \mathcal{G}(i,y) + \frac{\alpha}{K} \sum_{k \in \mathcal{I}_{\text{topk}}} \nabla_\theta \mathbf{z} \mathcal{G}(i,k) \tag{7}$$

From the equation above, we know that if we change $\mathbf{e}_y$ in the original $\gamma$ to $\tilde{\mathbf{e}}_{\text{topk}}$, the gradient of the new loss can be simplified to a combination of the above. Note that $A_{i,l} \cdot \text{sg}(\gamma_{i,l}) \cdot \nabla_\theta \mathbf{z} \mathcal{G}(i,k)$ is exactly the decomposition of $\gamma_{i,l}^{(k)} = \frac{\pi_\theta(y_{i,l}^{(k)}|s_{i,t})}{\pi_{\theta_{\text{ref}}}(y_{i,l}^{(k)}|s_{i,l})}$, i.e., updating the model using $y_{i,l}^{(k)}$. In other words, our new loss might have a form like

$$(1-\alpha)\gamma_{i,l} + \frac{\alpha}{K} \sum_{k \in \mathcal{I}_{\text{topk}}} \gamma_{i,l}^{(k)}$$

However, the combination above will make the new $\gamma$ a biased estimator (the RL theoretical guarantee needs a correct importance sampling). We then use the following stop-gradient trick to fix this. Specifically, by multiplying $\text{sg}\left(\gamma_{i,l}/\gamma_{i,l}^{(k)}\right)$ to the second term in Equation-(7), we can ensure that $\text{sg}(\gamma_{i,l}^{\text{pos}}) = \text{sg}(\gamma_{i,l})$. This design is only one line of code, using the `.detach()` in Pytorch.

## C  ADDITIONAL ANALYSIS OF TEARNING DYNAMICS

In Figure 8, we present the training dynamics for all three models, which validate our findings in Section 3 across multiple models.

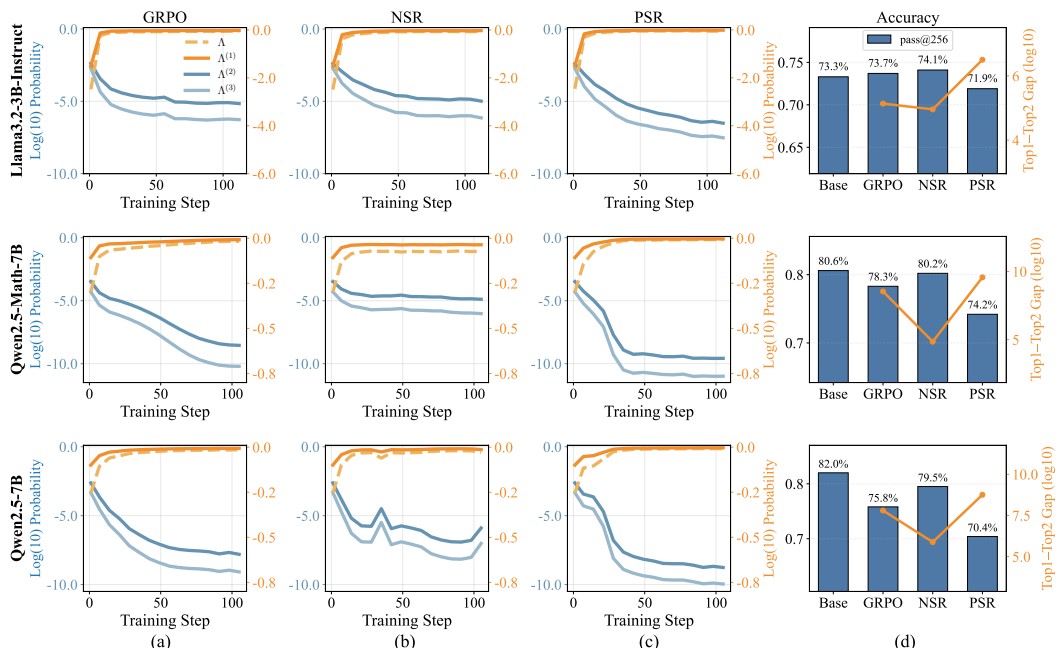

Figure 8: (a)-(c) Training dynamics of average log probability $\Lambda$ and top-K probabilities $\Lambda^{(k)}$ derived by GRPO, NSR, and PSR. (d) The corresponding pass@1 and pass@K results of the RLVL-trained models. Following the setups of Zhu et al. (2025), we train a Llama3.2-3B-Instruct on a mixture of GSM8K and MATH (Level 1) and train Qwen2.5-Math-7B on the MATH dataset.

## D  EXPERIMENT

### D.1  TRAINING DETAIL

### D.2  MORE EXPERIMENT RESULT

In this section, we present more detailed experimental results, as shown in Table 4.

## E  THE INFLUENCE OF SAMPLING TEMPERATURE

We evaluate Qwen2.5-Math-7B and its variants trained with GRPO and SimKO under different decoding temperatures ranging from 0.2 to 1.0. The results are shown in Table 5. Across all temperatures, SimKO consistently outperforms GRPO on both pass@1 and pass@K. Overall, pass@1 degrades as temperature increases, whereas pass@256 improves with higher temperatures.

Benefiting from the probability smoothing introduced on "forking" tokens, SimKO maintains strong pass@256 performance even at very low temperatures. For example, at temperature 0.2, SimKO achieves 77.2% in pass@256, compared to only 63.8% and 69.7% for Qwen2.5-Math-7B and GRPO, respectively. This indicates that SimKO enables the trained model to retain a broader solution paths. At higher temperatures, all methods become more exploratory, yet SimKO consistently preserves a clear advantage. These results demonstrate that our conclusions are not sensitive to a narrow temperature choice and that SimKO is robust to this decoding hyperparameter.

Table 4: Pass@1 / Pass@256 results for Qwen2.5-Math-7B, Qwen2.5-7B, and Llama3.2-3B-Instruct on MATH500, AIME 2024/25, Minerva_math, Olympiadbench, and AMC23 Datasets.

| Method | AIME24 | AIME25 | AMC23 | MATH500 | Minerva | Olympiad | Avg. |
|---|---|---|---|---|---|---|---|
| *Qwen2.5-Math-7B* | | | | | | | |
| Base Model | 13.2/66.0 | 5.4/51.8 | 38.2/98.5 | 55.8/96.0 | 16.5/68.8 | 25.6/77.0 | 25.8/76.4 |
| GRPO | 28.1/72.3 | 11.5/52.1 | 61.2/97.1 | 76.6/96.2 | 33.4/64.0 | 39.1/74.7 | 41.7/76.1 |
| PSR | 19.3/68.5 | 11.2/48.9 | 62.1/94.9 | 74.0/91.4 | 32.8/63.6 | 37.6/67.7 | 39.5/72.5 |
| NSR | 22.8/80.3 | 9.7/61.2 | 59.4/100.0 | 74.6/97.0 | 32.9/65.1 | 37.8/78.4 | 39.5/80.3 |
| W-REINFORCE | 29.2/86.5 | 10.8/55.7 | 61.1/97.4 | 76.4/96.4 | 33.4/67.6 | 38.1/77.6 | 41.5/80.2 |
| KL-Cov | 30.9/81.2 | 11.7/55.2 | 62.2/97.4 | 76.5/97.0 | 34.4/66.2 | 39.2/76.9 | 42.5/79.0 |
| GRPO w/Entropy-Adv | 29.1/81.7 | 10.9/55.0 | 62.5/92.1 | 77.1/95.0 | 33.5/60.7 | 39.7/71.9 | 42.1/76.1 |
| P@k T. | 26.7/77.9 | 10.2/61.3 | 58.8/97.5 | 73.3/96.8 | 33.2/68.8 | 36.6/78.2 | 39.8/80.1 |
| GRPO w/forking tokens | 28.6/74.6 | 11.5/57.4 | 59.6/96.7 | 77.4/94.4 | 33.9/65.1 | 39.6/72.4 | 41.8/76.8 |
| SimKO | 32.8/78.0 | 12.9/64.6 | 62.4/97.5 | 77.6/96.8 | 35.0/68.4 | 39.8/77.8 | 43.4/80.5 |
| Δ(SimKO-GRPO) | +4.7/+5.7 | +1.4/+12.5 | +1.2/+0.4 | +1.0/+0.6 | +1.6/+4.4 | +0.7/+3.1 | +1.7/+4.4 |
| *Qwen2.5-7B* | | | | | | | |
| Base Model | 7.4/64.0 | 3.6/48.8 | 36.1/99.6 | 61.4/97.2 | 23.0/70.2 | 28.1/78.5 | 26.6/76.4 |
| GRPO | 15.6/59.0 | 8.8/55.0 | 56.0/92.5 | 75.7/95.2 | 35.7/61.8 | 38.8/70.4 | 38.4/72.3 |
| PSR | 14.3/61.8 | 9.3/54.2 | 51.6/96.7 | 73.6/92.0 | 32.9/52.2 | 35.6/67.0 | 36.2/70.7 |
| NSR | 9.9/48.9 | 7.1/52.3 | 49.8/97.1 | 73.9/95.4 | 33.7/65.8 | 36.6/77.0 | 35.2/72.8 |
| W-REINFORCE | 11.2/57.5 | 6.1/54.2 | 54.3/99.6 | 73.9/95.8 | 33.4/66.2 | 36.6/77.9 | 35.9/75.2 |
| SimKO | 16.3/58.4 | 9.4/55.2 | 57.3/97.1 | 76.7/94.8 | 35.2/66.2 | 38.7/74.4 | 38.9/74.3 |
| Δ(SimKO-GRPO) | +0.7/-0.6 | +0.6/+0.2 | 1.3/+4.6 | 1.0/-0.4 | -0.5/+4.4 | -0.1/+4.0 | +0.5/+2.0 |
| *Llama3.2-3B-Instruct* | | | | | | | |
| Base Model | 3.4/51.7 | 0.7/46.7 | 20.3/94.9 | 37.8/93.6 | 10.1/59.2 | 12.7/67.1 | 14.2/68.9 |
| GRPO | 12.7/55.1 | 1.1/44.1 | 32.5/96.7 | 53.1/91.6 | 17.3/62.5 | 20.1/67.0 | 23.3/69.5 |
| PSR | 7.8/57.4 | 1.0/35.1 | 27.2/98.8 | 50.3/91.0 | 18.5/61.0 | 18.9/63.7 | 20.6/67.8 |
| NSR | 11.1/53.7 | 1.5/47.4 | 30.3/94.6 | 53.3/94.0 | 19.0/60.3 | 20.0/68.0 | 22.5/69.7 |
| W-REINFORCE | 13.3/51.7 | 1.1/42.1 | 31.4/96.3 | 52.4/92.8 | 16.7/59.9 | 19.6/65.8 | 22.4/68.1 |
| SimKO | 13.8/54.6 | 1.0/45.4 | 35.2/98.8 | 54.6/93.4 | 18.5/63.2 | 21.0/69.6 | 24.0/70.8 |
| Δ(SimKO-GRPO) | +1.1/-0.5 | -0.1/+1.3 | +2.7/+2.1 | +1.5/+1.8 | +1.2/+0.7 | +0.9/+2.6 | +0.7/+1.3 |

Table 5: Pass@1 / Pass@256 results for Qwen2.5-Math-7B across different temperatures on MATH500, AIME 2024/25, Minerva_math, Olympiadbench, and AMC23 Datasets.

| Method | Temp | AIME24 | AIME25 | AMC23 | MATH500 | Minerva | Olympiad | Avg. |
|---|---|---|---|---|---|---|---|---|
| Base Model | 0.2 | 13.3/55.0 | 5.3/41.9 | 40.7/82.5 | 62.8/93.2 | 14.8/44.1 | 28.2/66.2 | 27.5/63.8 |
| GRPO | 0.2 | 27.0/62.7 | 11.8/53.7 | 64.0/89.6 | 76.3/90.0 | 34.1/55.9 | 38.6/66.5 | 42.0/69.7 |
| SimKO | 0.2 | **31.9**/72.43 | **13.2**/57.5 | **63.9**/99.8 | **78.5**/96.8 | **36.2**/62.1 | **40.7**/74.7 | **44.1**/77.2 |
| Base Model | 0.4 | 13.8/58.0 | 5.5/50.9 | 42.4/91.7 | 61.1/95.4 | 15.6/61.8 | 27.6/73.5 | 27.7/71.9 |
| GRPO | 0.4 | 26.7/**75.6** | 12.2/54.6 | **63.4**/94.5 | 76.4/93.2 | 33.5/61.0 | 38.6/71.6 | 41.8/75.1 |
| SimKO | 0.4 | **31.1**/72.3 | **13.0/64.1** | 62.9/**97.5** | **78.1/97.4** | **36.1/66.2** | **40.6/76.1** | **43.6/78.9** |
| Base Model | 0.6 | 13.2/66.0 | 5.4/51.8 | 38.2/98.5 | 55.8/96.0 | 16.5/**68.8** | 25.6/**77.0** | 25.8/76.4 |
| GRPO | 0.6 | 26.5/75.1 | 12.3/52.3 | 62.9/98.9 | 76.5/93.8 | 33.9/63.6 | 38.5/73.8 | 41.8/76.3 |
| SimKO | 0.6 | **30.5/84.0** | **12.5/58.5** | **63.4/99.9** | **77.6/96.8** | **36.3**/66.9 | **40.0**/76.6 | **43.4/80.5** |
| Base Model | 0.8 | 26.2/75.5 | 4.2/48.5 | 34.8/**99.9** | 50.8/96.8 | 17.0/**69.9** | 23.1/**79.6** | 23.5/76.7 |
| GRPO | 0.8 | 26.6/**81.7** | 12.0/**52.3** | 61.9/97.4 | 76.5/95.2 | 33.5/64.0 | 38.5/74.5 | 41.5/77.5 |
| SimKO | 0.8 | **30.1**/78.9 | **12.5**/52.3 | **63.2**/99.5 | **77.0/97.2** | **35.5**/67.6 | **39.2**/77.5 | **42.9/78.8** |
| Base Model | 1.0 | 9.0/66.0 | 3.3/52.6 | 28.8/**99.6** | 42.8/**97.2** | 15.3/**71.7** | 19.0/**80.4** | 19.7/77.9 |
| GRPO | 1.0 | 26.2/75.5 | **11.7**/51.8 | 61.5/97.1 | **76.4**/95.6 | 33.5/63.2 | **38.3**/75.7 | 41.3/76.6 |
| SimKO | 1.0 | **28.9/75.6** | 11.0/**64.5** | **61.9**/97.4 | 76.0/96.6 | **34.8**/69.5 | 37.9/77.2 | **41.8/80.1** |

## F    THE USE OF LARGE LANGUAGE MODELS

We utilize large language models to assist in enhancing our writing by checking for typos, improving the fluency of expressions, and ensuring that the language adheres to academic standards. This helps make the writing more structured and professional, ultimately making it easier for the audience to understand the content.

