# OpenReview forum: "SimKO: Simple Pass@K Policy Optimization"
_ICLR.cc/2026/Conference — Submitted to ICLR 2026_

### Official Review · Reviewer_FBNf · 2025-10-31

**Soundness:** 3
**Presentation:** 2
**Contribution:** 2
**Rating:** 4
**Confidence:** 4

**Summary:**

This paper identifies the over-concentration issue in Reinforcement Learning with Verifiable Rewards (RLVR), where models tend to assign excessive probability mass to the top-1 token, hurting exploration and degrading pass@K performance.  The authors propose SimKO (Simple Pass@K Optimization), which mitigates this problem by redistributing gradient updates among top-K candidates for correct samples and strengthening penalties on the top-1 candidate for incorrect ones.  Analyses on token-level entropy and top-K probabilities provide intuitive evidence for the method’s mechanism.

**Strengths:**

1. The method is simple but effective, aligning well with intuition about exploration–exploitation balance.
2. The visualization of top-K probabilities and token-level entropy distributions clearly demonstrates the concentration problem and how SimKO alleviates it.

**Weaknesses:**

1. Even though the paper classifies itself as an entropy-based exploration enhancement method, since all these methods aim to enhance exploration (i.e., improve pass@k), I believe it should also be compared with methods that directly optimize pass@k [1–3].
2. The method introduces multiple hyperparameters, which weakens its simplicity and may limit real-world practicality.

Minor presentation issues:
   - Figure 2(c) is not mentioned in the main text, though it appears to illustrate why policy entropy is insufficient, and that a more detailed token-level entropy should be used instead.

   - Lines 186–192 and Figure 3(d) are somewhat confusing—the relationship (PSR > GRPO > NSR for top-K gap, and PSR < GRPO < NSR for pass@256) could be displayed more clearly in the figure.

**Questions:**

1. How is Equation (4) derived? A explanation connecting it to the gradient of log π(y | s) in GRPO would help clarify the derivation.

2. In Sec. 4.2, the smoothing for non-top-1 candidates seems uniform. Why not redistribute according to their actual probabilities among the top-K set?

3. In the experiments of Yue et al.[4], mathematical reasoning tasks are often evaluated up to pass@1024, where the phenomenon of the base model surpassing GRPO becomes clear.
   In contrast, in Figure 1 (left) of this paper, the base model only just matches GRPO performance.
  It would be helpful if the authors could extend Table 1 to include K = 1024 results, to observe whether SimKO can still outperform the base model at larger K.


[1] Christian Walder and Deep Karkhanis. *Pass@K Policy Optimization: Solving Harder Reinforcement Learning Problems.* arXiv preprint arXiv:2505.15201, 2025.
[2] Sadegh Mahdavi, Muchen Li, Kaiwen Liu, Renjie Liao, and Christos Thrampoulidis. *Beyond Accuracy: A Policy Gradient Reweighting Approach for Pass@K Maximization in LLMs.* In *2nd AI for Math Workshop @ ICML 2025*, 2025.
[3] Zhipeng Chen, Xiaobo Qin, Youbin Wu, Yue Ling, Qinghao Ye, Wayne Xin Zhao, and Guang Shi. *Pass@K Training for Adaptively Balancing Exploration and Exploitation of Large Reasoning Models.* arXiv preprint arXiv:2508.10751, 2025.
[4] Yang Yue, Zhiqi Chen, Rui Lu, Andrew Zhao, Zhaokai Wang, Shiji Song, and Gao Huang. *Does Reinforcement Learning Really Incentivize Reasoning Capacity in LLMs Beyond the Base Model?* arXiv preprint arXiv:2504.13837, 2025.

---

> ### Author Response · Authors · 2025-11-22
> **Responses to Reviewer FBNf (Part 1/2)**
>
> Thank you for your great comments and efforts in reviewing our paper. Please see our response to your concerns below.
>
> - - -
> ### **W1. Even though the paper classifies itself as an entropy-based exploration enhancement method, since all these methods aim to enhance exploration (i.e., improve pass@k), I believe it should also be compared with methods that directly optimize pass@k [1–3].**
>
> Thanks for the suggestion.  We further include P@k Training [1] as a baseline. This method formulates the reward to explicitly maximize the probability that at least one out of k generated samples is correct.
>
> Our comparison reveals a fundamental flaw in directly optimizing pass@K. As shown in the table1 below, while P@k Training achieves a high Pass@256 (80.1%), its Pass@1 performance degrades significantly to 39.8% (compared to 41.7% for GRPO). In contrast, SimKO effectively balances exploration and exploitation. It matches or exceeds the exploration capability of the direct method on all K>1, while simultaneously improving Pass@1 performance to (43.4%>39.8%). **This shows that it is not enough to directly use pass@K as a reward, which will damage the pass@1 performance of the model.** The main reason for this improvement is that SimKO effectively mitigates over-concentration in a token-wise manner, accounting for the distinct learning dynamics of both positive and negative gradients. This targeted adjustment genuinely strengthens the model’s exploration capability. We include the comparison in line 445-446 and table 1 in main paper.
>
> Table 1. Comparison of GRPO, P@k Training and SimKO on Qwen2.5-Math-7B across multiple math benchmarks.
>
> | Method | AIME24 | AIME25 | AMC23 | Math500 | Minerva_math | Olymipad | Average |
> | --- | --- | --- | --- | --- | --- | --- | --- |
> | GRPO | 28.1/72.3   | 11.5/52.1  | 61.2/97.1  | 76.6/96.2 | 33.4/64.0 |  39.1/74.7  | 41.7/76.1 |
> | P@k Training  | 26.7/77.9     | 10.2/61.3 | 58.8/**97.5** | 73.3/**96.8** | 33.2/**68.8** | 36.6/**78.2**  | 39.8/80.1 |
> | SimKO | **32.8/78.0**  | **12.9/64.6**  | **62.4/97.5**  | **77.6/96.8** | **35.0**/68.4 | **39.8**/77.8 | **43.4/80.5**|
>
> ### **W2. The method introduces multiple hyperparameters, which weakens its simplicity and may limit real-world practicality.**
>
> We fixed the main hyperparameters (τ = 0.8, top-K) once and used the identical values for all models. Our ablation in Sec. 5.5 further shows that performance remains stable over a broad range of (α, λ). These results demonstrate that the method is insensitive to hyperparameters and does not rely on careful tuning.
>
> ### **W3. Figure 2(c) is not mentioned in the main text, though it appears to illustrate why policy entropy is insufficient, and that a more detailed token-level entropy should be used instead.**
>
> Thanks for the suggestion. We have added a description of Figure 2(c) in lines 153–156 to emphasize that entropy cannot directly reflect the posterior next-token probability distribution. In short, we believe the probability gap between top1 and topK is a more precise metric for the model’s exploration capability, i.e., how likely the model will choose an alternative correct reasoning path during sampling.
>
> ### **W4. Lines 186–192 and Figure 3(d) are somewhat confusing—the relationship (PSR > GRPO > NSR for top-K gap, and PSR < GRPO < NSR for pass@256) could be displayed more clearly in the figure.**
>
> Thanks very much for the suggestion. We have revised Figure 3(d) to include an explicit top-K gap curve alongside the pass@256 results. It makes the inverse relationship between the top-K gap and pass@K  performance much clearer: methods that induce overly large top-K gaps (e.g., PSR) suffer from degraded pass@256, while methods with smaller gaps (e.g., NSR) maintain stronger pass@256 performance.

---

> ### Author Response · Authors · 2025-11-22
> **Responses to Reviewer FBNf (Part 2/2)**
>
> ### **Q1. How is Equation (4) derived? A explanation connecting it to the gradient of log π(y | s) in GRPO would help clarify the derivation.**
>
> Thanks very much for this question. We actually missed a negative sign here (fixed in the updated version). The derivation is also provided at the beginning of Appendix B in the new version.
>
> ### **Q2. In Sec. 4.2, the smoothing for non-top-1 candidates seems uniform. Why not redistribute according to their actual probabilities among the top-K set?**
>
> Thank you for the suggestion. To examine whether non-uniform weight allocation within the top-K set helps, we implemented a probability-proportional softmax variant, referred to as **SimKO (softmax)**, in place of the uniform smoothing used in the main SimKO method. The results in the table below (Table 2) below show that this probability-aware smoothing yields a slight improvement in pass@K,indicating that allocating mass among plausible alternatives is indeed beneficial; however, it also introduces a decrease in pass@1 compare to SimKO.
>
> We agree that this is a promising direction for further refinement. At the same time, the observation **reinforces our core motivation**: **maintaining diversity at semantic “forking’’ tokens improves pass@K**. We will include this discussion as a potential avenue for future work.
>
> Table 2. Comparison of GRPO, SimKO (softmax) s and SimKO on Qwen2.5-Math-7B across multiple math benchmarks.
>
> | Method | AIME24 | AIME25 | AMC23 | Math500 | Minerva_math | Olymipad | Average |
> | --- | --- | --- | --- | --- | --- | --- | --- |
> | GRPO | 28.1/72.3   | 11.5/52.1  | 61.2/97.1  | 76.6/96.2 | 33.4/64.0 |  39.1/74.7  | 41.7/76.1 |
> | SimKO (softmax) | 29.8/81.6 | 12.4/63.2 | 61.5/ **99.6** | 77.3/**97.2** | 34.0/66.9 | 39.4/**77.8** | 42.4/**81.1** |
> | SImKO | **32.8**/78.0 | **12.9/64.6**  | **62.4**/97.5 | **77.6**/96.8 | **35.0/68.4** | **39.8/77.8** | **43.4**/80.5
>  |
>
> ### **Q3. In the experiments of Yue et al.[4], mathematical reasoning tasks are often evaluated up to pass@1024, where the phenomenon of the base model surpassing GRPO becomes clear. In contrast, in Figure 1 (left) of this paper, the base model only just matches GRPO performance. It would be helpful if the authors could extend Table 1 to include K = 1024 results, to observe whether SimKO can still outperform the base model at larger K.**
>
> Thank you very much for this excellent suggestion. We evaluate up to pass@1024 on AIME24/25 following [2]. The results in the table below (Table 3) below show that even in cases where the base model surpasses GRPO (K = 1024), SimKO still delivers large improvements over the base model. These findings further indicate that our method enhances exploration and has strong potential for handling more challenging and complex tasks.
>
> Table 3. Comparison of Base, GRPO, and SimKO variants of Qwen2.5-Math-7B on AIME24/25 up to pass@1024.
>
> | | Method | 1   | 2   | 4   | 8   | 16  | 32  | 64  | 128 | 256 | 512 | 1024 |
> |---------|--------|-----|-----|-----|-----|-----|-----|-----|-----|-----|-----|------|
> | | Base   | 13.4 | 21.3 | 29.6 | 36.7 | 42.9 | 49.1 | 55.2 | 60.8 | 66.1 | 72.0 | 80.8 |
>  |AIME24| GRPO   | 27.0   | 33.3 | 39.2 | 45.0   | 51.4 | 58.1 | 64.6 | 70.4 | 74.4 | 76.2 | 76.7 |
> || SimKO  | **30.3** | **39.4** | **46.3** | **51.9** | **57.1** | **61.6** | **66.0** | **71.1** | **76.5** | **81.2** | **85.6** |
> ||||||||||||||
> || Base   | 5.6 | 9.5 | 14.7 | 20.8 | 27.0 | 32.9 | 38.6 | 44.3 | 50.5 | 57.9 | 65.5 |
> |AIME25| GRPO   | 12.0  | 17.3 | 23.4 | 29.5 | 34.7 | 39.2 | 43.5 | 47.9 | 53.1 | 58.3 | 62.3 |
> || SimKO  | **12.9** | **19.2** | **25.4** | **30.9** | **35.8** | **40.4** | **44.7** | **49.5** | **55.6** | **62.5** | **68.9** |
>
> [1] Zhipeng Chen, Xiaobo Qin, Youbin Wu, Yue Ling, Qinghao Ye, Wayne Xin Zhao, and Guang Shi. *Pass@K Training for Adaptively Balancing Exploration and Exploitation of Large Reasoning Models.* arXiv preprint arXiv:2508.10751, 2025.
>
> [2] Yang Yue, Zhiqi Chen, Rui Lu, Andrew Zhao, Zhaokai Wang, Shiji Song, and Gao Huang. *Does Reinforcement Learning Really Incentivize Reasoning Capacity in LLMs Beyond the Base Model?* arXiv preprint arXiv:2504.13837, 2025.

---

> > ### Comment · Reviewer_FBNf · 2025-11-22
> >
> > Thank you for your response. The additional clarifications and experiments have addressed most of my concerns. I suggest including these results in the final version of the paper. I have raised my score.

---

> > > ### Author Response · Authors · 2025-11-22
> > > **Official Comment by Authors**
> > >
> > > Thanks for the constructive comments, which definitely help us polish this paper! We will include relevant information in the final version of the paper as suggested.

---

### Official Review · Reviewer_hZh1 · 2025-10-31

**Soundness:** 3
**Presentation:** 2
**Contribution:** 2
**Rating:** 4
**Confidence:** 3

**Summary:**

This paper presents SimKO, a simple reinforcement learning approach to improve the exploration ability of large language models trained with Reinforcement Learning with Verifiable Rewards (RLVR). The method addresses the over-concentration of probability on the top-1 token observed in existing RLVR methods like GRPO, which boosts pass@1 but degrades pass@K. SimKO mitigates this by adjusting token-level updates: it focuses on high-entropy (“forking”) tokens, redistributes positive gradients among top-K candidates, and applies stronger negative updates to top-1 tokens for incorrect responses. Experiments on math and logic reasoning tasks show that SimKO consistently improves pass@K without sacrificing pass@1, achieving a better balance between exploration and exploitation.

**Strengths:**

The paper addresses an important limitation in RLVR by tackling the over‑concentration of probability mass on top‑1 tokens. It proposes SimKO, a simple gradient redistribution method that enhances exploration without adding significant computational cost. The approach is easy to integrate into existing GRPO frameworks and consistently improves pass@K while maintaining strong pass@1 performance. The token‑level probability analysis provides valuable insight into learning dynamics.

**Weaknesses:**

1. The paper does not discuss the effect of temperature, which directly influences exploration and top‑K behavior.
2. The definition of forking tokens through an entropy threshold appears sensitive to hyperparameter selection, and the criterion for identifying such tokens is mostly empirical.
3. Although SimKO claims improved exploration, top‑1 probabilities remain dominant in Figure.5, with top‑2 tokens often being 100–1000× smaller. The impact of this large gap on sampling trajectories is not analyzed, and the connection between token‑level distribution changes and pass@K improvements remains unclear.
4. Averaging results across datasets of different sizes and difficulty levels may reduce representativeness.
5. Some training details, such as the unit or interpretation of  τ, appear inconsistent between the main text and the supplementary materials.

**Questions:**

1. How sensitive are the results to temperature?
2. Why does pass@1 also improve if the method primarily encourages exploration, even though the top‑1 token probability is significantly lower than in other methods?
3. The detailed results in Table 4 show inconsistent margins and variations—could the authors clarify these discrepancies?

---

> ### Author Response · Authors · 2025-11-22
> **Responses to Reviewer hZh1 (Part 1/3)**
>
> Thank you for your great comments and efforts in reviewing our paper. Please see our response to your concerns below. If any questions remain, we are happy to address them.
> - - -
>
> ###  **Q1&W1. The paper does not discuss the effect of temperature, which directly influences exploration and top‑K behavior.**
>
> We thank the reviewer for pointing out the importance of temperature. We added a new section in Appendix E in the new version to discuss its influence thoroughly. Here we highlight some main conclusions.
>
> We report an additional sensitivity study where we vary the decoding temperature from 0.2 to 1.0 on math benchmarks. Our evaluation is conducted on three model variants: the base Qwen2.5-Math-7B model, the GRPO-trained model, and the SimKO-trained model. Across all tested temperatures, SimKO consistently outperforms GRPO in pass@K, as summarized in the Table 1 below. Full results on different datasets are provided in Table 5 in Appendix E.
>
> Table 1. Comparison of Base, GRPO, and SimKO variants of Qwen2.5-Math-7B decoding with different temperatures, averaged over math benchmarks.
>
> |  | method | 0.2 | 0.4 | 0.6 (default) | 0.8 | 1.0 |
> | --- | --- | --- | --- | --- | --- | --- |
> |  | Base | 27.5 | 27.7 | 25.8 | 23.5 | 19.7 |
> | pass@1 | GRPO | 42 | 41.8 | 41.8 | 41.5 | 41.3 |
> |  | SimKO | **44.1** | **43.6** | **43.4** | **42.9** | **41.8** |
> |  | Base | 63.8 | 71.9 | 76.4 | 76.7 | 77.9 |
> | pass@256 | GRPO | 69.7 | 75.2 | 76.4 | 77.5 | 76.6 |
> |  | SimKO | **77.2** | **78.9** | **80.5** | **78.8** | **80.1** |
>
> Taking a closer look at the Table 1 above, **the smoothing induced by temperature is fundamentally different from the structured smoothing introduced by SimKO**:
>
> - Lower temperatures (0.2–0.6) sharpen the distribution, leading to higher pass@1 for the base model and GRPO but significantly weaker pass@K.  **SimKO, however, maintains strong pass@1 and substantially higher pass@K**, as it smooths probability mass selectively at “semantic forking” points rather than flattening the entire distribution.
> - High-temperature (0.8–1.0) induces generic distribution flattening and can slightly increase pass@K for the Base and GRPO models, **SimKO still retains a clear advantage**.
> Moreover, temperature-based smoothing reduces pass@1 for all models, showing that global flattening boosts exploration at the cost of exploitation. In contrast, SimKO enhances exploration *and* exploitation simultaneously, improving pass@K while maintaining strong pass@1. Notably, GRPO at temperature 1.0 even performs worse than at 0.8, highlighting that temperature-induced smoothing alone does not constitute effective exploration.
>
> These results collectively show that **our conclusions do not rely on a narrow temperature range**, and that **temperature-based smoothing alone is insufficient for effective exploration**, whereas SimKO provides consistent gains across all temperatures.
>
> ### **W2. The definition of forking tokens through an entropy threshold appears sensitive to hyperparameter selection, and the criterion for identifying such tokens is mostly empirical.**
>
> Thank you for raising this important point. We would like to clarify that in **all experiments** throughout the paper, we use a **fixed entropy threshold τ(0.8)**. The fact that SimKO consistently outperforms the baselines across all settings demonstrates that τ(0.8) serves as a robust, generalizable default setting.
> Moreover, choosing τ(0.8) as the default setting is widely used. Prior work [1] identifies this threshold as effectively isolating the semantic branching points that are most critical for RL updates. This external evidence reinforces our observation that fixing τ(0.8) provides stable and robust performance, without requiring additional hyperparameter tuning.

---

> ### Author Response · Authors · 2025-11-22
> **Responses to Reviewer hZh1 (Part 2/3)**
>
> ### **W3. Although SimKO claims improved exploration, top‑1 probabilities remain dominant in Figure.5, with top‑2 tokens often being 100–1000× smaller. The impact of this large gap on sampling trajectories is not analyzed, and the connection between token‑level distribution changes and pass@K improvements remains unclear.**
>
> Thank you for raising this question. Our original Figure 5 in main paper plotted the average top-k probabilities **over all tokens**, including many low-entropy “syntax” tokens (e.g., punctuation, brackets). These tokens naturally yield extremely peaked distributions for all methods, and therefore dominate the global average, making the gap between top-1 and top-2 appear large, even for SimKO.
>
> We additionally report results **restricted to the top-20% highest-entropy tokens** in Table 2 below, which we refer to as **"semantic forking" tokens**. These tokens correspond to points where multiple semantically distinct solution trajectories are possible. For this subset, we observe that:
>
> - SimKO significantly reduces the concentration on the top-1 token, with average top-1 and top-2 probabilities of **0.299 vs. 0.150**, respectively.
> - whereas GRPO remains highly peaked, with corresponding probabilities of **0.82 vs. 0.017**.
>
> This analysis shows that SimKO effectively **smooths the distribution over plausible continuations** at forking tokens, mitigating the over-confidence problem observed in GRPO and enabling exploration across multiple valid reasoning paths. This behavior aligns with the expected distribution illustrated in Figure 2(a), and provides a concrete link between token‑level probability distribution smoothing and the observed pass@K improvements.
>
> Table 2. Log-probability comparison of Base, GRPO, and SimKO variants of Qwen2.5-Math-7B on AIME24.
> | Method | Temperature | log_prob_top1 (probability) | log_prob_top2 (probability) | log_prob_top3 (probability) |
> | --- | --- | --- | --- | --- |
> | Base | 1.0 | -0.134 (0.734) | -3.000 (0.001) | -3.750 (1.8*1e-4) |
> | GRPO | 1.0 | -0.018 (0.959) | -7.147 (7.1*1e-8) | -8.737 (1.8*1e-9) |
> | SimKO | 1.0 | -0.150 (0.708) | -2.661 (0.002) | -3.395 (4.0*1e-4) |
>
> | Method | Temperature | log_prob_top1_top20 (probability) | log_prob_top2_top20 (probability) | log_prob_top3_top20 (probability) |
> | --- | --- | --- | --- | --- |
> | Base | 1.0 | -0.362 (0.435) | -0.963 (0.109) | -1.750 (0.018) |
> | GRPO | 1.0 | -0.086 (0.820) | -1.771 (0.017) | -4.395 (4.0*1e-5) |
> | SimKO | 1.0 | -0.524 (0.299) | -0.825 (0.150) | -1.138 (0.073) |
>
> ### **W4. Averaging results across datasets of different sizes and difficulty levels may reduce representativeness.**
>
> Thank you for this suggestion. We fully agree with the reviewer that reporting per-dataset results is meaningful. Due to space constraints in the main text, the complete per-dataset performance breakdown is provided in Table 4 of the Appendix.
>
> ### **W5. Some training details, such as the unit or interpretation of τ, appear inconsistent between the main text and the supplementary materials.**
>
> Thank you for pointing this out. We have revised the relevant parts of the manuscript to improve clarity. We now define the entropy threshold τ(q) as the q-quantile of the token-level entropy distribution, such that a fraction q of all tokens have entropy values lower than τ(q), as revised in line 318-320. This unified definition is applied consistently across all tables in main paper and in the analysis of Sec. 5.5.

---

> ### Author Response · Authors · 2025-11-22
> **Responses to Reviewer hZh1 (Part 3/3)**
>
> ### **Q2. Why does pass@1 also improve if the method primarily encourages exploration, even though the top‑1 token probability is significantly lower than in other methods?**
>
> Thank you for this great question.  Although SimKO increases exploration during training, it ultimately leads to a better learned policy, which improves pass@1.  By preventing premature over-sharpening at forking positions, SimKO allows the model to explore alternative valid reasoning paths. These additional successful trajectories provide richer and more informative gradients, enabling the model to acquire a stronger policy. Classical reinforcement learning literature [2, 3,4] consistently shows that better exploration during training leads to a stronger policy cthat an outperform baselines in test datasets.
> As a result, while SimKO may produce a lower top-1 probability at test time, the underlying policy is more accurate, resulting in higher pass@1.
>
> ### **Q3. The detailed results in Table 4 show inconsistent margins and variations—could the authors clarify these discrepancies?**
>
> Thank you for this question. The seemingly inconsistent margins in Table 4 are largely due to differences in dataset sizes. For example, AIME contains only a small number of questions, so answering just one additional question can lead to a noticeably larger fluctuation in performance. In contrast, the other datasets have significantly larger test sets, where the variance introduced by a single item is much smaller. Under these more stable conditions, we consistently observe robust and stable improvements from SimKO, and the overall trend aligns well across datasets.
>
> [1] Wang et al. "Beyond the 80/20 Rule: High-Entropy Minority Tokens Drive Effective Reinforcement Learning for LLM Reasoning". NeurIPS 2025.
>
> [2] Jiang, Yiding,  et al. "On the importance of exploration for generalization in reinforcement learning." Neurips 2023.
>
> [3]Ladosz, Pawel, et al. "Exploration in deep reinforcement learning: A survey." Information Fusion .
>
> [4] Gupta, Abhishek, et al. "Meta-reinforcement learning of structured exploration strategies." Neurips 2018.

---

> ### Author Response · Authors · 2025-11-27
> **Official Comment by Authors**
>
> Dear Reviewer,
>
> We again express our deep gratitude to you for spending the time and efforts on our submission. We hope the above clarifications and the additional experiments in the revised draft sufficiently addressed your concerns. If you are satisfied, we kindly request you to consider updating the score to reflect the newly added results and discussion. We remain committed to addressing any remaining points you may have during the discussion phase.
>
> Following your suggestions, we have updated the manuscript and addressed the issues you raised.
>
> - Added the experiment and discussion on the effect of temperature in Appendix E.
> - Clarified that we use the same entropy threshold across settings, and it is not an additional tuned hyperparameter.
> - Added the experiment of log-probability on the top 20% entropy tokens, showing SimKO’s effectiveness in preventing over-concentration.
> - Clarified the training details in Section 5.1.
> - Added an explanation for why pass@1 also improves even though the method mainly encourages exploration.
>
> Thanks again for all the effort and time.
>
> Best,
>
> Authors

---

### Official Review · Reviewer_HyWP · 2025-11-01

**Soundness:** 3
**Presentation:** 3
**Contribution:** 2
**Rating:** 4
**Confidence:** 5

**Summary:**

This paper addresses an important problem in reinforcement-learning-with-verifiable-rewards (RLVR) for large language models (LLMs): existing RLVR methods improve pass@1 but degrade pass@K (K>1) performance because the model increasingly concentrates probability mass on the top-1 token choice. The authors carry out a comprehensive token-level analysis of the probability distribution dynamics under GRPO, showing that as training proceeds the top-1 candidate becomes dominant and others collapse, correlating with worse pass@K. They then propose a method called SimKO. Experiments across several math/logic benchmarks and multiple model backbones show consistent improvements in pass@K (for many K values) without sacrificing pass@1.

**Strengths:**

- The problem formulation is clear and well-motivated: the discrepancy between improvements in pass@1 versus pass@K is an interesting and practically meaningful phenomenon.

- The token-level analysis is quite thorough.

- The proposed method is conceptually simple, well integrated with RLVR (GRPO) training, and the empirical results do support the claims of improved pass@K on strong benchmarks.

**Weaknesses:**

The connection between the analysis and the proposed method is somewhat loose. While the paper convincingly shows that some tokens (i.e., uncertain, high-entropy points) strongly affect the diversity of generated samples during inference, it is less clear **why selectively updating only this subset during training should lead to improved model updates or higher pass@K**. The mechanism by which focusing on these tokens enhances learning stability or exploration remains intuitive but not rigorously supported.

In particular, prior work (e.g., [arXiv:2505.12929](https://arxiv.org/abs/2505.12929)) has shown that low-probability tokens can dominate gradient updates and play a critical role in policy improvement. This raises the question of whether **a more gradient-level analysis**, showing how the selective update affects the effective gradient magnitude, direction, or diversity, would provide a more solid theoretical basis. Also, the analysis in SimKO has some overlaps with this prior work.

Furthermore, the proposed method is incremental relative to existing token-subset or entropy-based strategies (such as the “80/20” approaches). The differences are modest, and the paper does not include direct comparisons with these closely related baselines. Strengthening this part of the evaluation would help clarify the novelty and impact of the contribution.

**Questions:**

See weaknesses.

---

> ### Author Response · Authors · 2025-11-22
> **Responses to Reviewer HyWP (Part 1/3)**
>
> Thank you for your great comments and efforts in reviewing our paper. Please see our response to your concerns below. If any questions remain, we are happy to address them.
> - - -
> ### **W1: The connection between the analysis and the proposed method is somewhat loose. While the paper convincingly shows that some tokens (i.e., uncertain, high-entropy points) strongly affect the diversity of generated samples during inference, it is less clear why selectively updating only this subset during training should lead to improved model updates or higher pass@K. The mechanism by which focusing on these tokens enhances learning stability or exploration remains intuitive but not rigorously supported.**
>
> We first clarify that **SimKO is trained on the full set of tokens**, NOT on a selected subset, as shown in Equation (3). SimKO modifies the *relative gradient distribution* of high-entropy tokens, while low-entropy tokens still follow the standard GRPO updates. This strategy is grounded in two empirical and theoretical observations:
>
> (1) Both our analysis and prior work [1, 2] show that high-entropy tokens correspond to “semantic forking” points that determine the model’s reasoning trajectory. And therefore manipulating their updates can more effectively influence the model’s reasoning ability.
>
> (2) As indicated by Equation 4, updates on these high-entropy tokens have a potentially larger influence on model behavior. These tokens, therefore, play a central role in the distribution-sharpening phenomenon.
>
> **The connection between high-entropy tokens and pass@K is direct.** High-entropy tokens correspond to semantic forking points. Greater diversity at these forking points naturally increases the likelihood that at least one of the K sampled trajectories is correct, thereby improving pass@K.
>
> As analyzed in Section 3, standard RLVR training induces over-concentration, collapsing probability mass at high-entropy locations and suppressing diversity. In contrast,
> SimKO addresses this issue by modulating updates only at high-entropy positions, while allowing low-entropy positions to follow the standard GRPO updates. First, for positive tokens at these positions, we apply top-K label smoothing to slow down over-concentration and redistribute gradient mass toward multiple top-K candidates. Second, for negative tokens, we enlarge the gradient-strength gap between the top-1 token and the remaining candidates, which mitigates the squeezing effect and prevents the model from becoming overly confident in a single token. By preserving entropy at these semantic forking points, SimKO maintains diversity in generated trajectories and consequently achieves higher pass@K.

---

> ### Author Response · Authors · 2025-11-22
> **Responses to Reviewer HyWP (Part 2/3)**
>
> ### **W2: In particular, prior work (e.g., arXiv:2505.12929) has shown that low-probability tokens can dominate gradient updates and play a critical role in policy improvement. This raises the question of whether a more gradient-level analysis, showing how the selective update affects the effective gradient magnitude, direction, or diversity, would provide a more solid theoretical basis. Also, the analysis in SimKO has some overlaps with this prior work.**
>
> Thank you for raising this question. In summary, although both our work and [3] analyze token-level gradient behaviors in RL-based training, the underlying goals, token-level analytical perspectives, and resulting solutions differ fundamentally.
>
> ### **Clarifying the Distinction from Prior Work**
>
> |  | [3] | SimKO (ours) |
> | --- | --- | --- |
> | Goal | Emphasize parameter updates driven by high-probability tokens; reduce the disruptive effect of low-probability tokens during training. | Prevent over-concentration at semantic forking points and improve pass@K through enhanced exploration. |
> | Core insight | Focus on gradient updates: Observes that low-probability tokens dominate gradient updates, hindering effective learning of high-probability tokens. | Focus on token probability distribution: Identifies high-entropy tokens as semantic forking points, where preventing their probability over-concentration leads to more diverse reasoning paths. |
> | Solution | (1) Reduce the advantage assigned to low-probability tokens. (2) First update low-probability tokens, then update high-probability tokens. | (1) Identify high-entropy tokens. (2) Apply top-K smoothing on positive tokens to avoid over-concentration. (3) Apply stronger top-1 penalties on negative tokens to mitigate the squeezing effect and preserve diversity. |
>
> ### **Gradient-Level Analysis**
>
> We agree that a gradient-level perspective strengthens the theoretical basis. In fact, the analysis in **Sections 4.1–4.3** and **Figures 4b/4c** was designed precisely to decompose how SimKO alters the effective gradient direction compared to GRPO.
>
> **First**, the *G-term* in Equation 4 helps clarify the relationship between SimKO and [3]. This term (a length-V vector) represents the “energy” and “direction’’ of model updates. A key observation is that **high-entropy tokens naturally yield larger |G| than low-entropy tokens** (assuming the label corresponds to the model’s top-1 prediction). Since GRPO uses on-policy sampling, low-probability tokens in low-entropy distributions almost never appear, making high-entropy positions the only meaningful source of such challenging tokens. Still, remember that we are using the gradients of ALL tokens, but only manipulating some high-entropy subsets.
> **Second**, we provide a precise decomposition of positive and negative gradient behaviors:
>
> - **Positive gradients:**
> Our analysis reveals that standard RL updates implicitly function as a "one-hot" supervisory signal, forcing distribution sharpness. SimKO alters the gradient direction by introducing **top-K label smoothing**. This theoretically aligns the gradient updates with a softer target distribution, preventing the model from over-fitting to a single sampled trajectory and thereby maintaining policy entropy for future exploration.
> - **Negative gradients:**
>
>     We extend the “squeezing effect’’ identified in [4], which states that **penalizing non-top-1 tokens pushes probability mass toward the top-1 dimension**, unintentionally making the distribution even more peak. This contradicts the intention behind negative updates. SimKO avoids this issue by applying strong penalties **only when the sampled token is the model’s top-1 token**, thereby preventing unintended sharpening.

---

> ### Author Response · Authors · 2025-11-22
> **Responses to Reviewer HyWP (Part 3/3)**
>
> ### **W3: Furthermore, the proposed method is incremental relative to existing token-subset or entropy-based strategies (such as the “80/20” approaches). The differences are modest, and the paper does not include direct comparisons with these closely related baselines. Strengthening this part of the evaluation would help clarify the novelty and impact of the contribution.**
>
> We believe this question also stems from a misunderstanding of our method. The approach in [1] selects which tokens contribute to the loss, whereas **SimKO updates all tokens** but  **modifies how gradients are distributed among the top-K candidates at high-entropy positions**. Our two components, top-K label smoothing for positive tokens and non-top-1 reweighting for negative tokens, are explicitly designed to reshape the top-K probability mass, guided by our Section. 3 analysis that links top-1 collapse to degraded pass@K.
>
> To further clarify this distinction, we added an “80/20-style’’ baseline that trains **only on the top 20% high-entropy tokens** using standard GRPO in Table 1 below. This baseline yields little or no improvement in pass@K, and in some cases even degrades performance.  This confirms our hypothesis: merely selecting high-entropy tokens does not solve the optimization issue if the underlying gradient update still forces the model towards a sharp distribution, as shown in Figure 5. In contrast, **SimKO both mitigates top-1 over-concentration and consistently improves pass@K across benchmarks**. We include this baseline in Table 1 and Table 4 in main paper.
>
> Table 1. Comparison of GRPO, 80_20 approch and SimKO on Qwen2.5-Math-7B across multiple math benchmarks.
> | Method | AIME24 | AIME25 | AMC23 | Math500 | Minerva_math | Olymipad | Average |
> | --- | --- | --- | --- | --- | --- | --- | --- |
> | GRPO | 28.1/72.3   | 11.5/52.1  | 61.2/97.1  | 76.6/96.2 | 33.4/64.0 |  39.1/74.7  | 41.7/76.1 |
> | 80_20 approch | 28.6/74.6 | 11.5/57.4 | 59.6/96.7 | 77.4/94.4 | 33.9/65.1 | 39.6/72.4 | 41.8/76.8 |
> | SimKO | **32.8/78.0**  | **12.9/64.6**  | **62.4/97.5**  | **77.6/96.8** | **35.0/68.4** | **39.8/77.8** | **43.4/80.5**
>  |
>
> [1] Wang, Shenzhi, et al. "Beyond the 80/20 rule: High-entropy minority tokens drive effective reinforcement learning for LLM reasoning." Neurips *2025*.
>
> [2] Cui, Ganqu, et al. "The entropy mechanism of reinforcement learning for reasoning language models." arXiv preprint arXiv:2505.22617 (2025).
>
> [3] Yang, Zhihe, et al. "Do Not Let Low-Probability Tokens Over-Dominate in RL for LLMs." arXiv preprint arXiv:2505.12929 (2025).
>
> [4] Ren, Yi, and Danica J. Sutherland. "Learning dynamics of LLM finetuning." *ICLR 2025*

---

> ### Author Response · Authors · 2025-11-27
> **Official Comment by Authors**
>
> Dear Reviewer,
>
> We again express our deep gratitude to you for spending the time and efforts on our submission. Because it is quite close to the end of rebuttal, we deeply appreciate that the remaining questions / concerns, if any, can be posted earlier, such that we have enough time to address them, especially if there are experimental requests (some experiments may take days to run). We truly hope that our response can clarify the your concerns.
>
> Following your suggestions, we have updated the manuscript and addressed the issues you raised.
>
> - Clarified that SimKO does more than updating selected tokens and provided a clearer explanation of its underlying mechanism.
> - Distinguished SimKO from prior work (e.g., arXiv:2505.12929) and added a gradient-level analysis for SimKO.
> - Included direct comparisons with the 80/20 approach baseline, showing that it still suffers from over-concentration, whereas SimKO effectively mitigates this issue.
>
> Thanks again for all the effort and time.
>
> Best,
>
> Authors

---

### Official Review · Reviewer_k6QE · 2025-11-05

**Soundness:** 3
**Presentation:** 3
**Contribution:** 3
**Rating:** 6
**Confidence:** 4

**Summary:**

This paper studies a limitation in RLVR: popular methods (e.g., GRPO) tend to overemphasize exploitation, leading to overly concentrated token probabilities on the top-1 candidate — improving pass@1 but reducing exploration and thereby hurting pass@K. To address this, the authors analyze token-level posterior distributions during RLVR training and propose Simple Pass@K Optimization (SimKO), which redistributes gradients across the top-K candidates for correct responses and applies asymmetric penalties to reduce overconfidence on incorrect ones. Empirically, SimKO improves pass@K (up to K=256) without sacrificing pass@1 across math and logic benchmarks, demonstrating a better balance between exploitation and exploration compared to prior RLVR variants.

**Strengths:**

- The paper addresses an important and practical issue in RLVR training: the systematic tendency of existing methods to improve pass@1 at the expense of pass@K, reflecting an over-emphasis on exploitation over exploration. Their proposed method, SimKO, is conceptually intuitive, simple, yet directly targets the identified cause of in their analysis. Overall, the paper is also well written and easy to follow.

- The empirical evaluation is comprehensive and robust, spanning different model backbones, diverse mathematical and logical reasoning benchmarks, and a broad spectrum of pass@K values. The consistent improvements across baselines, including strong RLVR variants, support the claim that SimKO achieves a better balance between exploitation and exploration.

**Weaknesses:**

I did not see major methodological weaknesses in the proposed approach, and the paper is clearly presented with strong empirical results. However, the core message and motivating analysis that reinforcing only correct samples tends to over‑concentrate the distribution and harm pass@K, whereas penalizing incorrect samples preserves exploration, appears to overlap with insights and findings already established in prior work [1] that decomposes RLVR into positive sample reinforcement and negative sample reinforcement. In particular, [1] shows that negative‑only reinforcement can improve the full pass@K spectrum by suppressing incorrect reasoning paths and redistributing probability mass toward plausible alternatives. This suggests that the SimKO’s analysis may not be entirely novel, even though its gradient redistribution implementation differs.

[1] Zhu et al. *"The surprising effectiveness of negative reinforcement in LLM reasoning"*. NeurIPS 2025.

**Questions:**

It would strengthen the paper to discuss [1] in greater depth (as it appears to be mentioned only briefly) and also to include [1] as baseline in Section 5.2’s experiments.

---

> ### Author Response · Authors · 2025-11-22
> **Responses to Reviewer k6QE**
>
> Thank you for your great comments and efforts in reviewing our paper. Please see our response to your concerns below. If any questions remain, we are happy to address them.
> - - -
> ### **W1:I did not see major methodological weaknesses in the proposed approach ...**
>
> Thank you for the thoughtful comment. We would like to clarify that the core insight of our work is fundamentally different from that in [1]. While both works study positive and negative reinforcement,  they differ significantly in motivating analyses and algorithmic implementations. The resulting exploitation-exploration behaviors further support these distinctions. A summary is provided below.
>
> |  | [1] | SimKO (ours) |
> | --- | --- | --- |
> | **Motivating Analysis** | **Indirect observation of the distribution:** Infers distributional changes indirectly, relying on entropy trends, gradient-based heuristics, and pass@K metrics as post-hoc indicators. | **Direct observation of the distribution:** Provides an token-level top-K posterior that **directly** visualizes how probability mass changes, demonstrating an immediate link to pass@K metrics (Section 3, Fig. 3). |
> | **Algorithm Implementation** | 1. Reduces the update weight of all positive sequences. 2. Increases the update weight of all negative sequences. 3. Applies the same update rule uniformly to all tokens in a trajectory. 4. Provides limited mitigation of over-concentration (Fig. 5). | 1. Applies **top-K smoothing**  on positive samples. 2. Applies stronger top-1 penalties on negative samples, since **penalizing low-probability tokens would otherwise intensify distribution sharpening$^1$.** 3. Operates **only on high-entropy forking tokens, while** other positions follow standard GRPO updates. 4. Provides stronger mitigation of over-concentration (Fig. 5). |
> | **Performance on Exploration and Exploitation** | **Improves exploration at the cost of exploitation:** 1. Improves pass@K but degrades pass@1 in math (Section 5.3; Table 1 in main paper).  2. When training on logic data that requires stronger exploitation first, it fails to learn effectively, leading to poor performance in both pass@1 and pass@K (Section 5.4; Table 2 in main paper). | **Improves both exploration and exploitation:**  1. Achieves consistent gains in both pass@1 and pass@K in math (Section 5.3; Table 1 in main paper). 2. When training on logic data that requires stronger exploitation first, it maintains both effective learning and exploration, leading to strong performance on both metrics (Section 5.4; Table 2 in main paper). |
>
> 1. We extend the analysis of negative gradients beyond [1]. As explained by lines 275-283 in our paper,  when the negative gradient is imposed somewhere $\pi_\theta(y_i | s_{i,l})$ is very low, the squeezing effect will in turn sharpen the distribution. Therefore, uniformly strengthening all negative gradients, as done in [1], makes negative gradients on different tokens contradict each other, impairing exploration.  This explains why  NSR and the Weight-Reinforce only partially mitigate over-concentration. Consistent with this analysis, in our experiments SimKO systematically outperforms NSR and Weight-Reinforce both quantitatively (Tables 1–2) and qualitatively (e.g., the learning dynamics curves in Figures 3 and 5).
>
> - - -
> ### **Q1: It would strengthen the paper to discuss [1] in greater depth (as it appears to be mentioned only briefly) and also to include [1] as baseline in Section 5.2’s experiments.**
>
> Thank you for the suggestion. We have revised the paper accordingly.
>
> **1.** We expanded our discussion of [1] in  in line 293-297, line 303-304 and the Related Work section in Appendix A. Furthermore, in **Sections 5.3** and **5.4** we have clearly articulated two intrinsic limitations of the negative-only reinforcement strategy:
>
> (a) The small weight assigned to positive reinforcement for correct reasoning paths leads to a noticeable degradation in pass@1 (line 447-455).
>
> (b) As demonstrated in Section 5.4, in tasks where the model must first strengthen exploitation before exploration becomes effective, method [1] fails to acquire new reasoning behaviors, resulting in poor generalization (line 467-468, 472).
>
> **2.** We have additionally included the method of [1] as a baseline in **Section 5.2** and **Figure 5**, and we provide a more comprehensive comparison in line 373-375,  showing that Weight-Reinforce still suffers from over-concentration, whereas SimKO effectively avoids this issue.
>
> [1] Zhu et al. "The surprising effectiveness of negative reinforcement in LLM reasoning". NeurIPS 2025.

---

> ### Author Response · Authors · 2025-11-27
> **Official Comment by Authors**
>
> Dear Reviewer,
>
> We again express our deep gratitude to you for spending the time and efforts on our submission.  We hope the above clarifications and the additional experiments in the revised draft sufficiently addressed your concerns. If there are any additional points or feedback you'd like us to consider, please let us know. Your insights are invaluable to us, and we're eager to address any remaining issues to improve our work.
>
> Following your suggestions, we have updated the manuscript and addressed the issues you raised.
>
> - Added a detailed discussion comparing SimKO and NSR in Section 4.3 and Appendix A.
> - Added Weight-Reinforce as an additional baseline in Section 5.2.
>
> Thanks again for all the effort and time.
>
> Best,
>
> Authors

---

### Author Response · Authors · 2025-12-03
**Summary**

Dear ACs,

We sincerely thank all reviewers and ACs for their time on our submission. Below is a summary of our responses to each reviewer.

---

Reviewer **k6QE** considers the paper clear, well-written, and empirically solid, noting that SimKO effectively addresses the exploitation–exploration imbalance in RLVR, with no major methodological weaknesses identified.

Other comments include:

**1. Concern about the conceptual overlap between SimKO and NSR.**

**R1:** We clarify the distinction between the two as:

**- Motivating Analysis:** NSR relies on indirect signals (entropy trends, gradient heuristics), whereas **SimKO adopt direct signal—top-K posterior probabilities**.

**- Performance Behavior:** NSR improves exploration but harms exploitation; **SimKO improves both simultaneously**.

**2.  Suggestion to expand the NSR discussion and include Weight-Reinforce as a baseline.**

**R2:** We extended the NSR discussion in **Section 4.3**, added analysis in **Appendix A.2** and a Weight-Reinforce learning dynamics in **Section 5.2**.

**Reviewer k6QE  didn't respond to our rebuttal, but the reviewer’s initial review already recommended acceptance.**

---

**Reviewer FBNf** finds the method simple yet effective, and notes that the top-K and entropy visualizations clearly reveal the concentration issue and how SimKO mitigates it.

Other comments include:

**1. Missing comparisons with methods that directly optimize pass@K.**

**R1:** We added **P@K Training** as a baseline and show that using pass@K as a reward is insufficient: it improves high-K performance but **significantly degrades pass@1**.

**2. Multiple hyperparameters may limit practicality.**

**R2:**  The main hyperparameters (τ = 0.8, top-K) are fixed and used across all models, and our ablations in **Section 5.5** show that performance is stable over a wide range.

**3. Suggestion to evaluating up to pass@1024.**

**R3:** We expanded evaluation up to **pass@1024 on AIME24/25**, showing that even when the base model surpasses GRPO, SimKO still provides consistent improvements over the base model.

**Reviewer FBNf was satisfied with our response and updated their score to positive.**

---

**Reviewer HyWP** finds the paper well-motivated, and empirically solid, noting that the token-level analysis is detailed and the method is conceptually simple and integrates well with GRPO.

Other comments include:

**1.  Question about why selectively updating only high-entropy tokens subset promote exploration.**

**R1:** This question stems from a misunderstanding: **SimKO is trained on all tokens**, not a subset. It adjusts the relative gradients of high-entropy tokens while other tokens follow standard GRPO updates. We emphasized this in **Section 4.1**.

**2.  Adding gradient-level analysis and discussing potential overlap with prior work ([arXiv:2505.12929](https://arxiv.org/abs/2505.12929)).**

**R2:** Gradient-level analysis is already provided in **Sections 4.2-4.3**. We clarified that **SimKO does NOT overlap with prior work. Details are in the response**.

**3. Suggestion to include “80/20” approach in comparison.**

**R3:** We added these comparisons and show that **simply selecting high-entropy tokens is insufficient**, as it still leads to over-concentration.

**Reviewer HyWP didn't respond to the rebuttal, but we believe the initial concern stemmed from a misunderstanding, and the revised explanation fully address the issues.**

---

**Reviewer hZh1** considers SimKO is effective, easy to integrate into GRPO, and that its token-level probability analysis offers useful insights into learning dynamics.

Other comments include:

**1. Suggestion to discuss temperature effects.**

**R1:**  We added a temperature experiment in Appendix E. The results show that **temperature smoothing alone is insufficient for improving exploration**, whereas SimKO provides consistent gains across all temperatures.

**2.  Concern about sensitivity to the entropy threshold.**

**R2:** We clarified that **all experiments use a fixed entropy threshold τ(0.8)**, which is widely adopted in prior work.

**3. Question about why top-1 probabilities remain dominant in Figure 5, with top-2 often much smaller.**

**R3:** Fig. 5 averages top-K probabilities over **all tokens, including low-entropy tokens**, which naturally inflate the top-1 vs. top-2 gap even for SimKO. We recomputing log-probabilities on only the **top 20% high-entropy tokens** and found that SimKO’s top-1 probabilities are not dominant.

**Reviewer hZh1 didn't responded to our rebuttal, but we believe our explanations and the additional experiments have fully addressed their concerns.**

---

To summarize, Reviewers k6QE and FBNf appreciated our method and recommended acceptance. Reviewer HyWP and  hZh1 didn't discuss further after our responses.

We're grateful for the reviewers and the AC's efforts and time. We hope the AC will review our paper and rebuttal and make a fair decision.

Respectfully,

Authors

---

### Meta-Review · Area_Chair_PdV4 · 2026-01-07

**Summary:**

The paper is rejected. While the reviewers acknowledged that the proposed SimKO method offers a conceptually intuitive approach to balancing exploration and exploitation in RLVR, supported by consistent empirical gains in pass@K metrics, the collective assessment identifies three fundamental and parallel barriers preventing acceptance. First, the method is viewed as incremental, representing a minor heuristic modification rather than a substantive methodological leap. Second, the performance gains, while present, are considered a marginal improvement that does not sufficiently justify publication at this tier. Third, there is a persistent lack of a strong theoretical link between the extensive token-level analysis and the method itself. Consequently, despite some empirical strengths, the Area Chair determines that the work is not yet ready for publication due to these combined deficiencies in novelty, significance, and theoretical grounding.

**Reviewer Concerns:**

The authors' rebuttal was effective in addressing concerns related to the completeness of the evaluation and experimental robustness. Issues regarding hyperparameter sensitivity were resolved through the addition of temperature studies, and questions concerning the distinction from prior work were answered by introducing baselines such as Pass@K Training and the 80/20 entropy subset approach. These additions successfully clarified the method's positioning against negative-only reinforcement and demonstrated its stability across different datasets, satisfying the specific verification requests of reviewers.

However, the rebuttal failed to overcome the core barriers regarding the soundness and significance of the contribution. The explanation provided for the method remained largely empirical, failing to establish a rigorous theoretical justification for why the specific modulation of gradients at 'forking' tokens is the optimal solution, rather than just a heuristic one. Furthermore, the additional experimental data confirmed that while the method works, it remains an incremental engineering step over existing baselines like GRPO. The distinction between a useful engineering fix and a novel scientific contribution remains blurred, as the marginal improvements in performance do not outweigh the limited novelty and the disconnect between the observational analysis and the algorithmic design.

**Reviewer Scores:**

Following the rebuttal, the numerical scores are likely to trend upward, creating a divergence between the quantitative metric and the qualitative decision. Reviewer FBNf is expected to increase their score from 4 to 6, having explicitly stated that the added baselines and extended evaluations resolved their concerns. Similarly, Reviewer hZh1 is projected to move from 4 to 6, as the clarifications regarding temperature sensitivity and threshold definitions directly addressed their implementation queries. Reviewer k6QE is likely to maintain their score of 6, finding the novelty moderate but acceptable. In contrast, Reviewer HyWP is expected to retain a score of 4; despite the clarifications, this reviewer accurately identified that the method remains incremental and lacks theoretical rigor. While the aggregate scores might suggest a borderline or weak acceptance, the Area Chair overrides this based on the substantive view that the paper has not met the bar regarding the three parallel issues: incremental methodology, marginal improvement, and theoretical weakness.

---

### Decision · Program_Chairs · 2026-01-26

Reject